# Investigation of biometabolites and novel antimicrobial peptides derived from promising source *Cordyceps militaris* and effect of non-small cell lung cancer genes computationally

**Muhammad Afzal[1]\*, Mai Abdel Haleem A. Abusalah[2], Neelum Shehzadi[1], Muhammad Absar[3], Naveed Ahmed[3], Sarmir Khan[4], Yalnaz Naseem[5], Noshaba Mehmood[1], Kirnpal Kaur Banga Singh[3]\***

1 Faculty of Science and Technology, Department of Basic and Applied Chemistry, University of Central Punjab, Lahore, Pakistan, 2 Faculty of Allied Medical Sciences, Department of Medical Laboratory Sciences, Al-Ahliyya Amman University, Amman, Jordan, 3 Department of Medical Microbiology and Parasitology, School of Medical Sciences, Universiti Sains Malaysia, Kelantan, Malaysia, 4 Department of Pharmacy-Drug Sciences, University of Bari Aldo Moro, Bari, Italy, 5 Institute of Drug Discovery and Development, Zhengzhou University, Zhengzhou, China

\* dr.afzal@ucp.edu.pk (MA); kiren@usm.my (KKBS)

## Abstract

Mushrooms are considered one of the safe and effective medications because they have great economic importance due to countless biological properties. *Cordyceps militaris* contains bioactive compounds with antioxidant, antimicrobial and anti-cancerous properties. This study was projected to analyze the potentials of biometabolites and to extract antimicrobial peptides and protein from the *C. militaris*. An *in-vitro* analysis of biometabolites and antimicrobial peptides was performed to investigate their pharmacological potentials followed by quantification and characterization of extracted protein. Computational analysis on non-small cell lung cancer genes (NSCLC) was performed on quantified compounds to interpret the biometabolites from *C. militaris* that could be potential drug candidate molecules with high specificity and potency. A total of 34 compounds representing 100% of total detected constituents identified were identified using GCMS analysis and 20 compounds using LC-MS which showed strong biological activities. FT-IR spectroscopy manifest powerful instant peaks to have different bioactive components including carboxylic acid, phenols, amines and alkanes present in methanolic extract of *C. militaris*. In *C. militaris*, higher protein concentration was observed in 70% concentration of protein extract (500 µg/ml ± 0.025). The best antioxidant activity (% Radical scavenging activity) of methanolic extracts was 80a ± 0.03, antidiabetic activity was 37 ± 0.057 and anti-inflammatory activity was 40 ± 0.021 at 12 mg/ml. Antibacterial activity for different concentrations of Cordyceps protein and methanolic extracts was significantly ($p < 0.05$). Indolizine, 2-(4-methylphenyl) has most binding affinity (micromolar) and optimized properties to be selected as the lead inhibitor. It interacts favorably with the active site of RET gene of NSCLC and is neuroprotective and hepatoprotective.

**Data Availability Statement:** All relevant data are within the manuscript and its Supporting Information files.

**Funding:** The author(s) received no specific funding for this work.

## Introduction

Mushrooms are widely distributed throughout the earth biosphere [1]. Mushrooms are among the best substitute sources secondary metabolites includes anthraquinones, terpenes, steroids, derivatives of benzoic acid, as well as contains few primary metabolites with proteins, peptides, and oxalic acid [2]. Mushrooms possess various pharmacological importance including antibacterial, radical scavenging, immunomodulating, antitumor, anti-hypercholesterolemic, cardiovascular, antifungal, antidiabetic, hepatoprotective, anti-parasitic and detoxification effects [3–5].

*Cordyceps militaris* is the complex composed of the fruiting body (the grass part), and the sclerotium (the dead body of insect), with the source of biologically active compounds that studied in nutraceuticals, biotechnology and therapeutic applications [6] and with the functions of improving, immunity, anti-inflammatory, antibacterial, blood pressure and lowering blood glucose [7]. Different bioactive compounds which includes polysaccharides, adenosine derivatives (mostly cordycepin), mannitol, ergosterol, protein and peptides are present in fruiting bodies and mycelia of *C. militaris* [8]. The *C. militaris* show variety of therapeutic properties which includes anti-inflammatory and antioxidant properties, reduce physical fatigue, help to manage blood glucose and to treat chronic kidney disease, improves liver functions and boost physical performance, regulation of blood pressure and beneficial uptake of oxygen in the body and also have antitumor and antiaging properties [9].

Antioxidant is a molecule that inhibits the oxidation process. When a chain reaction takes place in the cell, it causes death or damage to the cell [10]. Antioxidants stop these chain reactions in a cell by removing free radicals and inhibiting different oxidation reactions [11]. The *C. militaris* contains bioactive compounds such as polysaccharides, phenolic compounds, flavonoids, and cordycepin, which have been shown to possess antioxidant activity. These compounds can scavenge free radicals and reduce oxidative damage to cells and tissues [12].

The antidiabetic activity is also exhibited by the *C. militaris* [13]. The main source of glucose in the blood is the hydrolysis of starch. In small intestines, only monosaccharides such as fructose and glucose can absorb from our diet, so polysaccharides in our diet need to be broken down into monosaccharides [14]. The α-amylase enzyme hydrolyzes the oligosaccharides linkage of α-1,4-glycosidic and further breaks down disaccharides into monosaccharides (simple sugar) [15]. The α-glucosidase generates glucose from the non-reducing ends of oligosaccharides [16]. The main inhibitor of α-glucosidase is acarbose which is commonly used to decrease glucose production by inhibiting this enzyme in small intestines [17]. Studies have demonstrated that *C. militaris* extracts can stimulate glucose uptake in skeletal muscle cells and adipocytes by activating glucose transporters such as GLUT4. By increasing glucose uptake into cells, *C. militaris* may help lower blood glucose levels and improve overall metabolic health [18,19].

The *C. militaris* has shown promising effects on the regulation of genes involved in non-small cell lung cancer (NSCLC) [20,21]. Studies have indicated that *C. militaris* can modulate the expression of key oncogenes and tumor suppressor genes, leading to the inhibition of cancer cell proliferation, induction of apoptosis, and reduction of metastasis [22]. The bioactive components of *C. militaris* have been observed to downregulate genes associated with cell cycle progression and angiogenesis, while upregulating those involved in apoptosis pathways, such as the activation of caspases and the regulation of Bcl-2 family proteins [23,24]. Additionally, *C. militaris* has been reported to influence the expression of genes related to immune responses, potentially enhancing the body's ability to fight against NSCLC [25]. These gene-modulating effects position *C. militaris* as a potential adjunct therapy for NSCLC, offering a natural approach to complement conventional treatments and improve patient outcomes [26,27].

Most of the studies have been conducted on the other species of Cordyceps rather than *C. militaris* [28]. To the best of our knowledge, there is no current literature reported on GC-MS and LC-MS interpretation of methanolic extract of *C. militaris* and isolation of antimicrobial peptides by treatment with proteolytic enzyme. The present work was planned to identify and characterize biological active compounds from fraction based on polarities of methanolic crude extract of *C. militaris* and to isolate the mushroom bioactive peptides (MBAPs) from the *C. militaris*. Different protocols and different strains were used to check out the activity of bio-metabolites and peptides against bacteria and with the DPPH, α-amylase and BSA assay to find out the antioxidant, anti-diabetic and anti-inflammatory effect respectively and the characterization by SDS-PAGE. In this study we investigated biometabolites and antimicrobial peptides from medicinal mushroom to eliminate the risks of microbial infections and other ailments. Computation analysis on non-small cell lung cancer genes (NSCLC) was performed by GC-MS quantified compound to evaluate the biometabolites from *C. militaris*.

## Materials and methods

### Sample collection

The fresh fruiting bodies of medicinal mushroom *C. militaris* was purchased from the house of Mushroom, Lahore. The mushroom was taxonomically classified, identified and confirmed from the University of Central Punjab, Lahore Pakistan.

### Preparation of methanol extract of *Cordyceps militaris*

**Sample preparation.** Fresh fruiting bodies of *C. militaris* was rinsed with distilled water to detach the contamination. The dried fruiting bodies of *C. militaris* were taken in a pestle and mortal. By adding liquid nitrogen, the fruiting bodies of mushroom was crushed till it turns into powder form by lysis its cell wall. Finally, the dried sample was crushed into fine powder.

**Extraction method.** The mushroom sample was soaked in methanol and this suspension was incubated in shaking incubator for the duration of 10 days. After the assigned period, the sample mixture was filtered through cheese cloth and then further by Whatman filter paper. At 50˚C, on rotary evaporator, filtrate was further concentrated. The procedure was repeated until the desired semi-solid sample is obtained. The sample was collected and well-kept-up in air tight glass tubes. At the end, the methanol extract of fruiting bodies of *C. militaris* were ready for further analysis of the biological activities.

**Fourier Transform Infrared Spectroscopy (FT-IR).** The methanol extract of *C. militaris* were mixed up in KBr salts and converted to fine pellet using a pestle and mortal. For spectroscopy analysis, two FT-IR spectrometers with 4000 and 500 cm$^{-1}$ scan range were used.

**Gas chromatography-mass spectrometry.** GC-MS (Model: 7890B, 5977A, Agilent USA) was used to evaluate the whole methanolic extract of the *C. militaris*. A DB-5 capillary tube with 30 m length, 0.25 μm film thickness and 0.25 mm diameter was used to investigate the biometabolites of methanolic extract using a Hewlett-Packard model 6890 working ionization energy at 75 eV. The carrier gas was helium, which was used a rate of 1 ml/minute. MS source was set at temperature to 280˚C, the split ratio to 1:6, injected 1 μL of sample extract down the column. In order to interpret and analyze the components after separation in the column at 75 eV, FID spectroscopy was used. In NIST library, the spectra of the unknown components to known components were compared.

**Liquid chromatography-mass spectrometry.** LC-ESI-MS which constitutes the triple quadrupole liquid chromatography-mass spectrometer (Finnigan TSQ Quantum Ultra EMR, Thermo Scientific) was used to analyze the methanolic extract of *C. militaris*. Components can

be separated by Liquid Chromatography and charge ions identified by the Mass Spectrometer (MS). The chemicals were separated chromatographically using a Shim-pack XR-ODS III column. The mass spectrometer was set to Multiple Reaction Monitoring mode, and the fragmentation patterns of the standards were analyzed by injecting each standard solution at 0.1 g/mL directly into mass spectrometer. By comparing mass spectrometry fragmentation patterns and their retention time to those of legitimate standards, the major components of *C. militaris* were discovered.

## Biological activities

**Antioxidant activity (DPPH).** The radical scavenging of DPPH was done by the procedure given by [29] with modification. Different concentration of extracts and ascorbic acid (12, 6, 3, 1.5, 0.75 mg/mL) of 1 mL were mixed with 1 mL of 90 µM DPPH in dark. It was then incubated for 30 minutes at room temperature. Positive control was ascorbic acid while DPPH with methanol was used as negative control. Triplicates were run for the result at different time intervals.

$$\% \text{ Radical Scavenging activity } = \frac{\text{Blank absorbance} - \text{Sample absorbance}}{\text{Blank absorbance}} \times 100$$

**Antibacterial activity.** Antibacterial activity of methanol extract of *C. militaris* was investigated by disc diffusion method in which 4 bacterial strains, two Gram-positive bacteria (*Streptococcus viridans* and *Staphylococcus aureus*) and two Gram-negative bacteria (*Klebsiella pneumoniae* and *Escherichia coli*) were used. Nutrient broth was used for culture preparation. Hundred ml of nutrient formed by dissolving 1 g of peptone, 1 g NaCl and 0.5 g yeast mixed in 100 ml of distilled water in a flask and autoclave at 121˚C for 1 hour. The mixture was poured into different test tubes and 100 µl of bacterial strains is added to each test tube and labeled accordingly. It was then incubated for 24 hours in a shaking incubator [30].

**Disc diffusion method.** Microorganism suspension was prepared and was evenly distributed on a solid Muller Hinton agar media plate using sterilized swab sticks. A solution was made from each strain that was equal to 0.5 McFarland standards. Sterilized discs were dipped in methanol extract of 200 mg/ml concentration and placed on a media plate. The plates were wrapped tightly with parafilm and placed on an incubator for 24 hours at 37˚C. Positive control was Cefotaxime while negative control was DMSO. The zone of inhibition was calculated as mm [31].

**Antidiabetic activity (DNSA method).** *In-vitro* antidiabetic activity was performed using DNSA method by protocol given by [32]. Different concentrations (12, 6, 3, 1.5, 0.75 mg/ml) were devised. Take 5 µl of α–amylase was treated with 195 µl of methanolic extract of *C. militaris*. Incubate the sample at 37˚C for 10 minutes. After incubation, add 50 µl of 1% starch solution, incubate for 10 minutes. Then added 50 µl of DNSA solution and further incubate it for 10 minutes. Absorbance was checked by ELISA at 630 nm. Negative control is distilled water while metformin exploited as a standard. Test was performed in triplicate at different time intervals.

$$\% \text{ Inhibition } = \frac{\text{Control absorbance} - \text{Sample absorbance}}{\text{Control absorbance}} \times 100$$

**Anti-inflammatory activity.** The effect of sample extract against anti-inflammatory activity was checked by the method reported by [33] with little modifications. Different

concentration of sample extracts of 12, 6, 3, 1.5, 0.75 mg/ml were devised and 100 μl of sample extract was taken in centrifuge tubes and add 0.5 ml of BSA. The mixture was incubated for 20 minutes at 37˚C and then retained on a water bath for 10 minutes at 70˚C. Take absorbance reading at ELISA reader at 630 nm in Elisa plates. The % BSA inhibition was intended by the formula given below.

$$\% \text{ BSA Inhibition } = \frac{\text{Control absorbance} - \text{Sample absorbance}}{\text{Control absorbance}} \times 100$$

**Extraction of peptides.** About 50 g of dried fruiting bodies of *C. militaris* were treated with liquid nitrogen and squashed in chilled pestle and mortal. Phosphate buffer saline (PBS) was added into the crushed fruiting bodies after a minute with pH 7.4. Freeze the sample at -20˚C for further liquification and procedure was repeated for another cycle [34]. Sample hence obtained was centrifuged at 10000 rpm for 15 to 20 minutes and segregate the supernatant. Supernatant was stored at -20˚C for next procedure.

**Salting in and salting out.** The ammonium sulfate treatment was used to precipitate out the supernatant. Thirty, 50, and 70% ammonium sulfate precipitation were used for salting-in [35]. To remove salt, we poured the sample onto a dialysis membrane, sealed it on both sides, and set it on a magnetic stirrer for 12 hours. Afterwards, the sample was spun at 6,000 rpm for ten to fifteen minutes. The resulting pellet was stored in PBS buffer at -20˚C for future use [36].

## Protein quantification and characterization

**Bradford assay.** To quantify the unknown protein concentration, at first a standard curve was generated using BSA as a standard and making its dilutions for obtaining different concentrations at different levels. Stock solution of BSA is prepared that was 1 mg/mL and then dilutions of 10, 20, 40, 60, 80, 100 μl were devised. The absorbance was measured using an ELISA reader. In each well of the 96-well Elisa plate, add 250 μl of Bradford reagent and 10 μl of either the BSA standard or the unknown sample fractions. Every sample and standard was run three times to determine the absorbance at 630 nm. More intense the blue color, more concentration of protein is present. In order to determine the total protein content of the unknown sample, the absorbance readings were compared to the standard curve [37].

**Characterization of protein.** The samples were prepared for the SDS-PAGE to characterized the protein. The Protein ladder used for SDS-PAGE was Thermo Fisher Scientific prestained protein ladder. The protein range was 10–180 kDa which was the mixture of ten prestained recombinant proteins with different chromophores [38].

## Biological activities for Cordyceps protein

**Antimicrobial activity.** Antimicrobial assay of mushroom sample was performed. Two strains of bacteria were used gram positive bacteria *Staphylococcus aureus* and *Streptococcus viridans* and Gram-negative bacteria *Escherichia coli* and *Klebsiella pneumoniae*.

Nutrient broth was used for culture preparation. 100 mL of nutrient formed by dissolving 1 g of peptone, 1 g NaCl and 0.5 g yeast mixed in 100 mL of distilled water in a flask and autoclave d at 121˚C for 1 hour. The mixture was poured into different test tubes and 100 μL of bacterial strains is added to each test tube and labeled accordingly. It was then incubated for 24 hours in a shaking incubator.

A conical flask containing 250 ml of distilled water and 10 g of disintegrated nutritional agar was autoclaved at 121˚C for 1 hour in order to prepare agar plates. The agar mixture was then added to autoclaved plates for a few minutes.

**Disc diffusion method.**  The culture medium was poured onto the petri plates and labelled in biosafety cabinet. Using the culture swab, the bacterial culture was spread onto petri plates. Antibiotic discs (filter paper) were dipped into 100 µl of protein extract and placed on a media plate. The plates were wrapped tightly with parafilm and placed in an incubator for 24 hours at 37°C for bacterial growth. Partial peptides having antimicrobial activity showed clear zone of inhibition. For positive control as Cefotaxime was used while PBS taken as negative control.

## Insilico analysis

**Ligand retrieval.**  A methanolic crude extract of *C. millitaris* was recovered as a ligand by GC-MS. The chemicals were isolated and studied. 3D structure of all compounds was obtained from PubChem database.

**Target protein selection.**  Targeted susceptible protein was identified from the literature. 3D structure of targeted protein was retrieved from the PDB database (https://www.rcsb.org) (Accessed on 1 September 2023). 3D structure of NSCLC gene of RET (PDB ID: 6NEC), PIK3CA (PDB ID: 7L1B), and TCTN3 (PDB ID: 7QRX) was downloaded from PDB database.

**Molecular docking.**  AutoDoc Vina was used to performed docking for RET (PDB ID: 6NEC) and phytocompounds (ligands) of *C. militaris* through GC-MS analysis. PubChem (http://pubchem.ncbi.nlm.nih.gov/) accessed on 3 September 2023 download ligands in SDF format. 3D molecular structure of protein with unique PDB ID was downloaded in PDB format from Protein Data Bank (PDB). While, discovery studio software (Version 2021) was used to prepare the target protein [39].

**Lead optimization.**  Lead optimization was used to enhance the ligand's binding affinity, pharmacokinetic properties, selectivity and safety profiles. It was performed through the Bio-SolveIT software suite. Energy minimization was performed in UCSF Chimera (version 1.17.1) using the Gasteiger approach to get a zero net charge on the ligands. Following the optimization process, the ligands that were chosen underwent a conversion into a 'mol2' file format in order to facilitate molecular docking and subsequent evaluation.

**Pharmacokinetics properties.**  SwissADME was used to characterize the highest binding energy compound against NSCLC genes. It was used to evaluate the pharmacokinetics, drug-likeness of the drug, lipophilicity, along with its solubility, and toxicity [40].

**Toxicity screening (ProTox-II).**  For predicting various toxicity endpoints, such as hepatotoxicity, carcinotoxicity, immunotoxicity, cytotoxicity, toxicity targets and their adverse outcome pathways, Protox-II was used.

**Molecular dynamic simulation.**  For conducting high-performance molecular dynamics simulations, GROMACS version 2023 which is accessed at the web address: https://www. gromacs.org/. It operates by simulating the depiction of intricate domain motions within macroscopic molecules. It was used to investigate various ligand structural conformations residing inside the binding pocket while keeping fixed particular position of other molecules or amino acids or conformational fluctuations that may occur during molecular simulations and gives the stability insights of interaction between protein and ligand.

**MMPBSA analysis.**  The Molecular Mechanics Poisson Boltzmann Surface Area (MMPBSA) was manipulated in screening step for post-refinement and predicted free binding energy calculation $G_{binding}$ by using GROMACS version 2023. The outcomes derived from the MMPBSA computations encompassed Binding Free Energy (BFE) values. The results are widely acknowledging as very precise outcomes in the realm of computational analysis for protein-ligand complexes simulated insilico. In this analysis, electrostatic forces, van der walls interactions, molecular mechanics potential energy and free energy of solvation, including polar and non-polar interactions were involved in the analysis of binding free energy (BFE).

MMPBSA methods used the following equations in order to interpretation of protein and ligand binding free energy by using GROMACS version 2023.

i. $\Delta G_{bind} = G_{complex} - (G_{protein} + G_{ligand})$

ii. $G_x = <E_{MM}> - TS + <G_{solvation}>$

iii. $E_{MM} = E_{bonded} + E_{non\text{-}bonded} = E_{bonded} + (E_{vdw} + E_{elec})$

iv. $G_{solvation} = G_{polar} + G_{non\text{-}polar}$

v. $G_{non\text{-}polar} = \gamma SASA + b$

Where in Eq (i), $G_{complex}$ is the whole complex free energy while the $G_{protein}$ and $G_{ligand}$ are the protein and ligand's total free energy respectively. In Eq (ii) and (iii) $G_x$ is the free energy for the individual species, $E_{MM}$ is the vacuum molecular mechanics' potential energy and it is the sum of bonded, electrostatic interaction and van der Waals potentials. TS shows the binding free energy entropic contribution, where T is the temperature and S indicates the entropy, and $G_{solvation}$ depicts the solvation free energy for both polar and non-polar solvation free energy. In Eq (iv) $G_{polar}$ demonstrated the electrostatic free energy solvation while $G_{non\text{-}polar}$ is the non-electrostatic free energy solvation. In Eq (v) SASA indicates the solvent-accessible surface area while $\gamma$ and $b$ are the empirical constant.

## Statistical analysis

Each result is expressed as a mean and standard error of measurement. The difference between treated group and control group was compared using one-way ANOVA followed by Dunnett's post hoc test. The analysis was carried out by using SPPSS and significant $p > 0.05$ were considered.

## Results

### Fourier transform infrared spectroscopy

FTIR spectroscopy manifest to have different bioactive components, including carboxylic acid, phenols, amines and alkanes present in methanolic extract of *C. militaris*. Eight functional groups identified from the methanol extract are shown in **Table 1** and **Fig 1**. The powerful instance peak with the O-H stretching at 3323 cm$^{-1}$ were observed.

### Gas chromatography- mass spectrometry

GC-MS analysis shows the chemical composition of methanolic extract of *C. militaris* in **S1 Fig**. Thirty-four compounds show 100% of total detected constituents were identified and the

**Table 1. FTIR interception of components of methanol extract of *C. militaris*.**

| Sr. no. | Wavenumber cm$^{-1}$ | Functional group | Compound Class |
|---|---|---|---|
| 1 | 3323.22 | O-H stretching | Alcohols, phenols |
| 2 | 2944.82 | C-H stretching | Alkane |
| 3 | 2833.03 | C-H stretching | Alkane |
| 4 | 1655.20 | N-H bending | Amine |
| 5 | 1449.05 | O-H bending | Carboxylic acid |
| 6 | 1112.57 | C-N stretching | Amine |
| 7 | 1020.50 | C-N stretching | Amine |
| 8 | 611.07 | C-H bending | Alkane |

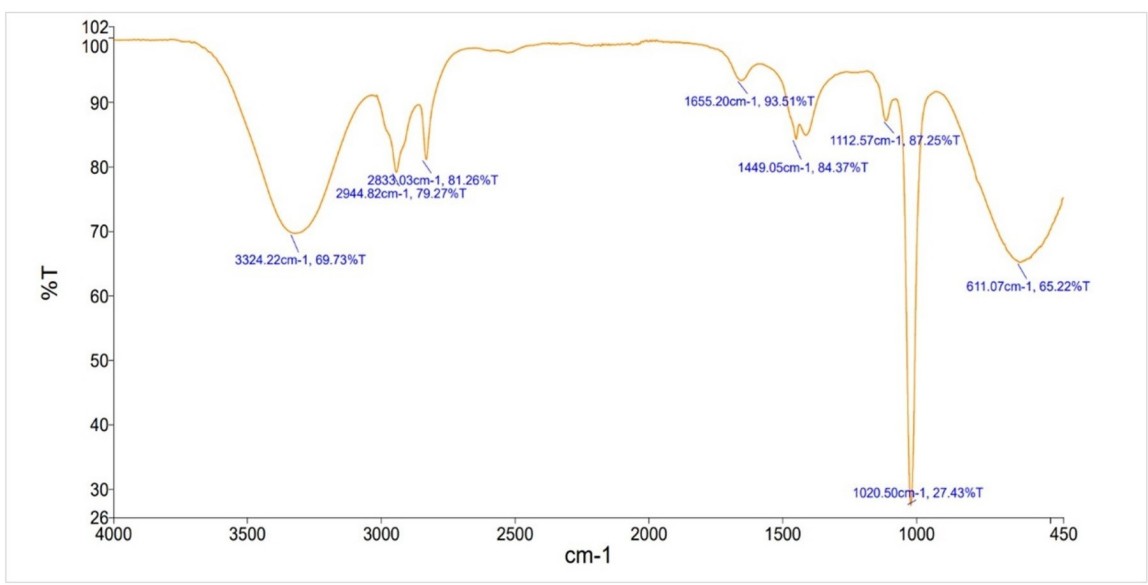

**Fig 1. FTIR analysis of methanol extract of *C. militaris*.**

GC-MS analysis. The major compounds were 1,2,3-Benzenetriol (21.77), 5-Hydroxymethyl-furfural (19.61), Methyl 3,4,5-trihydroxybenzoate (5.45), 2(5H)-Furanone (3.70), and 2-Furan-carboxylic acid (2.65) shown in **S1 Table**.

## Liquid chromatography-mass spectrometry

Twenty compounds representing 100% of total detected constituents were identified and the major compounds were malonic acid, 1,2,3-Trihydroxybenzene, 5-Ethoxy-4,5-dihydro-2(3H)-furanone, Caftaric acid, 4-Ethoxy-4-oxobutanoic, Oxoadipic acid, Trans-Aconitic acid, 3-Methyl-4-hydroxy-5-all-trans-heptaprenylbenzoate and Ampeloside Bs1 as shown in **S2 Table**.

## Biological activities of methanolic extract of *Cordyceps mlitaris*

**Antioxidant activity.** The antioxidant activity of *Cordyceps militaris* of methanolic extract fractions at different concentrations (12, 6, 3, 1.5, 0.75 mg/ml) were assessed by DPPH radical scavenging assay. The result indicated that methanolic extract of *Cordyceps militaris* show a high percentage of inhibition (80 ± 0.03) at 12 mg/ml concentration and lowest (55 ± 0.09) at 0.75 mg/ml concentration as manifested in **Table 2**. The free radical scavenging ability of *C. militaris* varies remarkably ($p < 0.005$).

**Antimicrobial activity.** The results obtained for antibacterial activity are shown in **Table 3**. In *Escherichia coli*, methanol extract manifests the maximum zone of inhibition (6.9 ± 0.04). While (6.5 ± 0.01), (3 ± 0.98) and (3.2 ± 0.42) zone of inhibition against bacterial strains *Klebsiella pneumonia*, *Streptococcus viridans*, and *Staphylococcus aureus* respectively (**S2 Fig**). Positive control was Cefotaxime. While negative control was DMSO. The ability of inhibition of bacterial strains of methanol extract of *C. militaris* varies significantly ($p < 0.05$).

**Antidiabetic activity.** The antidiabetic activity of methanolic extract of *Cordyceps militaris* at different concentrations (12, 6, 3, 1.5, 0.75 mg/ml) was analyzed as shown in **Table 4**. The highest antidiabetic activity was shown (37 ± 0.057) at 12 mg/ml and lowest activity

**Table 2. DPPH percentage radical scavenging activity of methanol extract of *C. militaris*.**

| Fractions | Concentrations (mg/mL) | DPPH % Inhibition (Mean ± S.D.) | IC$_{50}$ |
|---|---|---|---|
| Methanol | 0.75 | 55 ± 0.09 | 2.71 |
| | 1.5 | 60 ± 0.05 | |
| | 3 | 61 ± 0.02 | |
| | 6 | 76 ± 0.04 | |
| | 12 | 80 ± 0.03 | |
| Ascorbic acid | 0.75 | 80 ± 0.07 | 3.89 |
| | 1.5 | 83 ± 0.06 | |
| | 3 | 85 ± 0.03 | |
| | 6 | 89 ± 0.02 | |

The values were the average of triplicate samples (n = 3) ± S.D., the results are significant because one-way ANOVA shows (p ≤ 0.05).

(13 ± 0.084) at 0.75 mg/ml. The percentage inhibition of antidiabetic activity of *C. militaris* is significantly p < 0.05.

**Anti-inflammatory activity.** To assess the anti-inflammatory mechanism of *C. militaris*, its methanolic extract capability to inhibit bovine serum denaturation is calculated at different concentrations (12, 6, 3, 1.5, 0.75 mg/ml) shown in **Table 5**. It was observed that methanolic extract shows maximum inhibition of denaturation of BSA (40 ± 0.021) which is less than standard diclofenac (86 ± 0.096) at 12 μl/ml concentration. The result showed a concentration-dependent on inhibition of protein denaturation by *C. militaris*. The ability to protein inhibition (BSA) of *C. militaris* varies significantly (p < 0.05).

## Protein quantification

**Bradford method.** Total protein quantification was done by Bradford assay. Unknown protein concentration sample was found out by comparing the sample absorbance with the standard curve. Different concentrations of BSA were used and its standard curve was obtained as shown in **Fig 2A**.

Protein concentration calculated from Bradford assay was summarized as the 70% concentration showed higher protein quantity which was 500 μg/ml. While 30% and 50% showed the 250 μg/ml and 400 μg/ml respectively, as manifested in **Table 6**.

**Table 3. Antibacterial activity of methanol extract of *C. militaris* against bacterial strains.**

| Bacterial strains | Fractions (mg/ml) | Zone of inhibition (mm) Mean ± S.D |
|---|---|---|
| *Escherichia coli* | Methanol | 6.9 ± 0.04 |
| | Cefotaxime | 11 ± 0.85 |
| *Klebsiella pneumoniae* | Methanol | 6.5 ± 0.01 |
| | Cefotaxime | 10.2 ± 0.34 |
| *Streptococcus viridans* | Methanol | 3 ± 0.98 |
| | Cefotaxime | 6 ± 0.34 |
| *Staphylococcus aureus* | Methanol | 3.2 ± 0.42 |
| | Cefotaxime | 11.4 ± 0.65 |

The values were the average of triplicate samples (n = 3) ± S.D., the results are significant because one-way ANOVA shows (p < 0.05).

**Table 4. Antidiabetic activity of methanol extract of *C. militaris*.**

| Fractions | Concentrations (mg/mL) | % Inhibition of α-Amylase (Mean ± S.D) | IC$_{50}$ |
|---|---|---|---|
| Methanol | 0.75 | 13 ± 0.084 | 4.26 |
| | 1.5 | 19 ± 0.089 | |
| | 3 | 21 ± 0.046 | |
| | 6 | 34 ± 0.023 | |
| | 12 | 37 ± 0.057 | |
| Ascorbic acid | 0.75 | 34 ± 0.029 | 1.17 |
| | 1.5 | 38 ± 0.087 | |
| | 3 | 42 ± 0.056 | |
| | 6 | 45 ± 0.019 | |

The values were the average of triplicate samples (n = 3) ± S.D., the results are significant because one-way ANOVA shows (p ≤ 0.05).

## Characterization of protein

**SDS-PAGE.** The sample were subjected to SDS-PAGE to find out the molecular weight of the protein as shown in **Fig 3A**. Thermo Fisher Scientific pre-stained Protein Ladder was used which ranging from ~10 to ~180 kDa manifested in **Fig 3B**. The sample with 70% shows the maximum bands compared to the 50 and 30% ranges from ~100 to ~10 kDa. 70% showed cluster of bands with size of 70 kDa, 64 kDa, 59 kDa, 45 kDa, 32 kDa, 22 kDa. 50% showed weak bands with the size of 40 kDa, and 22 kDa. while 30% showed no band.

**Proteolytic enzyme treatment.** Protein isolated from *Cordyceps militaris* treated with proteolytic enzyme (i.e., pepsin, trypsin and papain) in order to make the peptides depicted in **Fig 3C**. The purpose of using proteolytic enzymes to check whether Cordyceps protein of higher kDa digested to convert into lower kDa antimicrobial peptides. The optimum temperature and pH for pepsin is 37˚C and 2–4 respectively [41] and for trypsin the optimum temperature and pH is 65˚C and 9 respectively [42]. For papain, the optimum temperature and pH is 37-80˚C and 7 respectively [43]. Thermo Fisher Scientific pre-stained Protein Ladder was used ranging from ~10 to ~180 kDa. SDS-PAGE analysis of the proteolytic enzymes treated extract showed only trypsin digestion and show maximum peptides in lower kDa between 15–10 kDa.

**Table 5. Anti-inflammatory activity of methanol extract of *C. militaris*.**

| Fractions | Concentrations (mg/mL) | BSA inhibition (Mean ± S.D.) | IC$_{50}$ |
|---|---|---|---|
| Methanol | 0.75 | 23 ± 0.097 | 1.75 |
| | 1.5 | 27 ± 0.070 | |
| | 3 | 34 ± 0.035 | |
| | 6 | 38 ± 0.089 | |
| | 12 | 40 ± 0.021 | |
| Ascorbic acid | 0.75 | 45 ± 0.096 | 1.48 |
| | 1.5 | 48 ± 0.091 | |
| | 3 | 53 ± 0.098 | |
| | 6 | 80 ± 0.012 | |

The values were the average of triplicate samples (n = 3) ± S.D., the results are significant because one-way ANOVA shows (p ≤ 0.05).

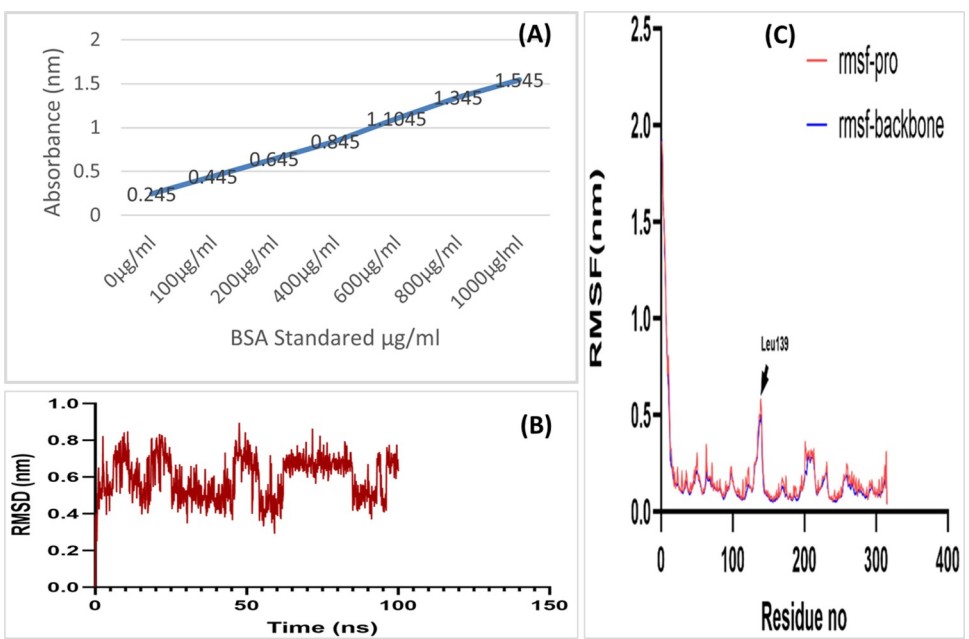

**Fig 2.** (A) BSA Standard Curve by Bradford assay. (B) RMSD stability study of Indolizine, 2-(4-methylphenyl). This manifest the RMSD in 'nm' of Indolizine, 2-(4-methylphenyl) following its alignment to the Time (ns). These findings determined the extent to which Indolizine, 2-(4-methylphenyl) binds to the protein (C) Detection of RMSF with protein backbone.

## Biological activities of *Cordyceps militaris* protein

**Anti-bacterial activity.** Anti-bacterial activity was performed for identifying the peptides activity from the fruiting bodies of *C. militaris* at different concentrations and zone of inhibition measured in mm, as shown in **Table 7**. Fractions of 70%, 50% and 30% showed visible zone of inhibition i.e., (4 ± 0.22), (3.5 ± 0.06) and (3.6 ± 0.01) in *Escherichia coli* respectively. In *Klebsiella pneumonia*, 70%, 50%, and 30% fraction showed the (2.7 ± 0.06), (0.5 ± 0.09) and (0.2 ± 0.03) zone of inhibition respectively.

The highest antibacterial activity found against *Staphylococcus aureus* which showed the maximum zone of inhibition in 70% i.e., (5.4 ± 0.08), while 50% and 30% was (3 ± 0.02) and (1 ± 0.04) respectively. In *Streptococcus viridans*, 70%, 50% and 30% fraction showed the (4.3 ± 0.034), (3.3 ± 0.07) and (4 ± 0.176) zone of inhibition respectively. Positive control is Cefotaxime. Negative control is PBS. The ability of inhibition of bacterial strains of protein fractions of *C. militaris* varies significantly ($p < 0.05$).

## Effect of GC-MS Quantified Phytoconstituents on Non-Small Cell Lung Cancer (NSCLC) genes

**Protein retrieval.** 3D structure of targeted protein of NSCLC i.e., RET gene (PDB ID: 6NEC), PIK3CA (PDB ID: 7L1B), and TCTN3 gene (PDB ID: 7QRX) was downloaded from Protein DataBank (PDB).

**Table 6. Total protein concentration in different fractions of *C. militaris*.**

| Sr. no. | Fraction (%) | Concentration (µg/ml) |
|---|---|---|
| 1 | 30 | 250 µg/ml ± 0.08 |
| 2 | 50 | 400 µg/ml ± 0.010 |
| 3 | 70 | 500 µg/ml ± 0.030 |

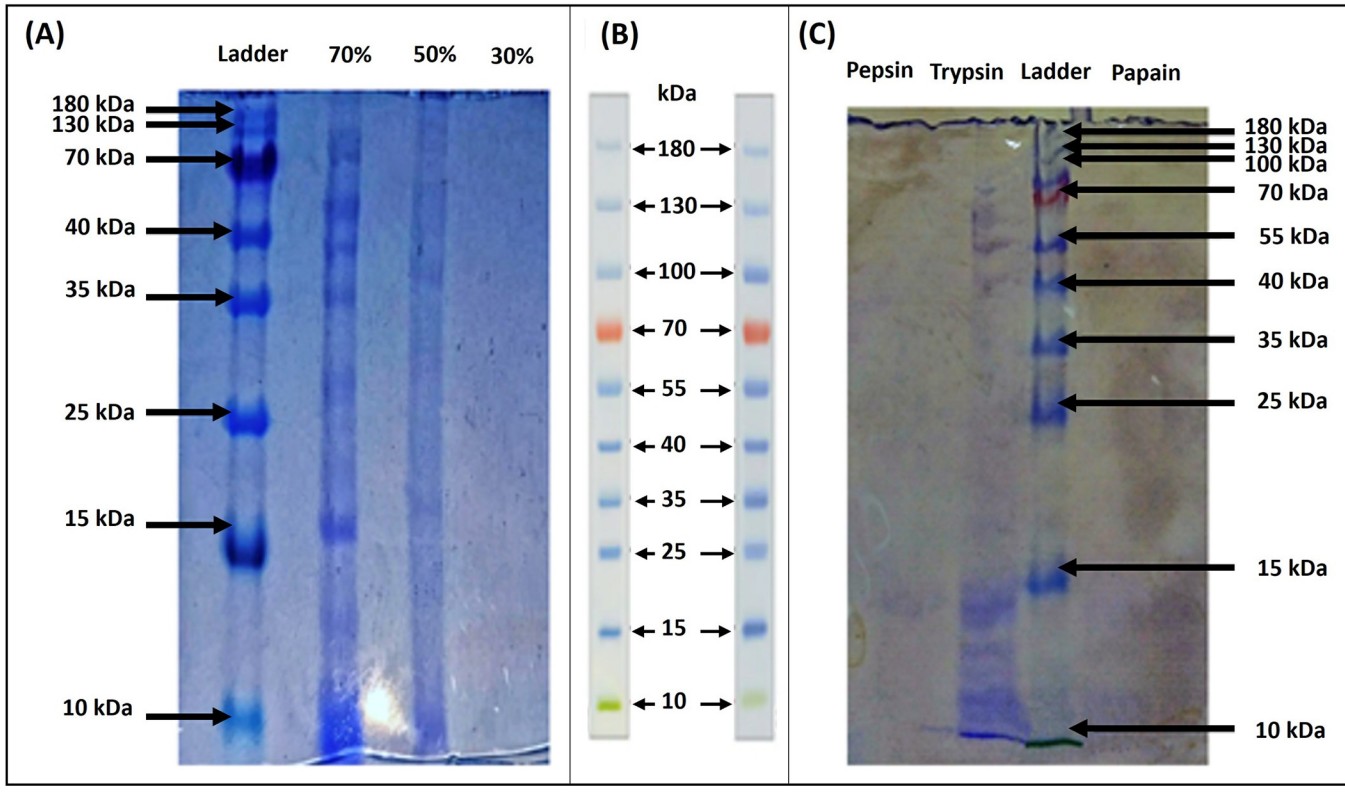

**Fig 3. SDS-PAGE analysis.** (A) SDS-PAGE of *Cordyceps militaris* protein (B) Thermo Scientific ᵀᴹPageRulerᵀᴹ pre-stained Protein Ladder size (kDa) (C) Proteolytic enzyme treatment of Cordyceps protein.

**Table 7. Antibacterial activity of *C. militaris*.**

| Bacterial strains | Fractions | Zone of inhibition (mm) Mean ± S.D |
|---|---|---|
| *Escherichia coli* | 30% | 3.6 ± 0.01 |
| | 50% | 3.5 ± 0.06 |
| | 70% | 4 ± 0.22 |
| | Cefotaxime | 11 ± 0.09 |
| *Klebsiella pneumoniae* | 30% | 0.2 ± 0.03 |
| | 50% | 0.5 ± 0.09 |
| | 70% | 2.7 ± 0.06 |
| | Cefotaxime | 9.4 ± 0.07 |
| *Streptococcus viridans* | 30% | 4 ± 0.176 |
| | 50% | 3.3 ± 0.07 |
| | 70% | 4.3 ± 0.034 |
| | Cefotaxime | 6 ± 0.10 |
| *Staphylococcus aureus* | 30% | 1 ± 0.04 |
| | 50% | 3 ± 0.02 |
| | 70% | 5.4 ± 0.08 |
| | Cefotaxime | 6 ± 0.07 |

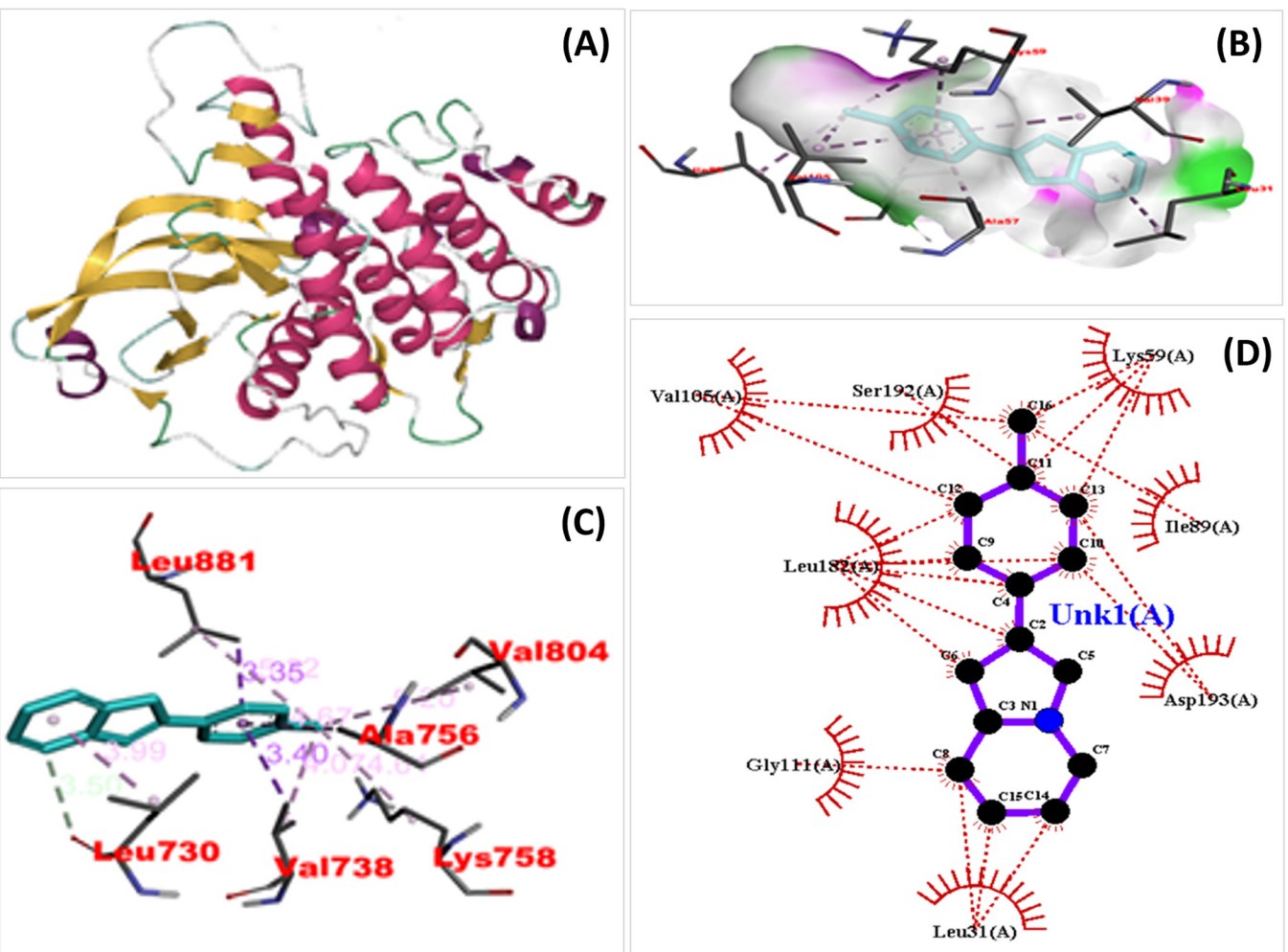

**Fig 4. Molecular interactions of Indolizine, 2-(4-methylphenyl)- and RET.** (A) Binding pockets of RET protein by DeepSite. Active residues are highlighted by yellow arrows (B) H-bond showing 3-dimensional image of docked complex (C) 3D image of docked complex showing amino acids and distance of protein-ligand interaction (D) Visualization of best pose of Indolizine, 2-(4-methylphenyl) in complex with RET gene.

**Active sites prediction.** DeepSite tool were performed to predict the binding pockets of RET gene (PDB ID: 6NEC) which involve in causing NSCLC disease. Five main binding sites were identified and pockets highlighted by yellow arrows as shown in **Fig 4A**. The binding positions and their score are shown in **S3 Table** respectively.

**Molecular docking.** AutoDoc Vina was used which determined the binding energies and the highest binding scores obtained by Indolizine, 2-(4-methylphenyl)- in NSCLC at RET gene (PDB ID: 6NEC) while second highest binding energy in PIK3CA (PDB ID: 7L1B), and TCTN3 gene (PDB ID: 7QRX) as compared to RET gene as shown in **S4 Table**. Fig 4B and 4C reveal that the compounds' binding interactions were mostly caused by alkyl forces in the evaluated compounds and pi-interactions between the ligand and non-small cell lung cancer protein.

## Molecular dynamic simulation

**Molecular dynamic simulation (100ns).** The molecular dynamics simulations for 100ns were performed with GROMACS version 2023, an open-source suite specifically intended for

high-performance molecular dynamics simulations. The software can be accessed at the following web address: https://www.gromacs.org/.This particular service functions by replicating the representation of complex domain movements within macroscopic molecules. Moreover, it facilitates the exploration of potential structural alterations or conformational variations that may occur during molecular simulations, while providing valuable insights into protein-ligand interactions stability.

Molecular docking results indicate that among the several compounds examined, 2-(4-methylphenyl) Indolizine exhibited a superior dock score, accompanied by the highest cluster. The co-crystallized structure obtained from the X-ray structure is utilized for conducting a molecular dynamics (MD) simulation investigation. The dock score (-CDOCKER interaction energy, kcal/mol) of Indolizine, 2-(4-methylphenyl) was -8.0 kcal/mol with RET gene. The pose with the highest score within this cluster was selected as the optimal pose and subsequently subjected to Molecular Dynamics Simulations (MDS) in order to evaluate its stability for the whole duration of the simulation run as depicted in **Fig 4D**. The purpose of conducting Molecular Dynamics Simulations (MDS) was to get a deeper understanding of the behavior and movement of the ligand within the protein's binding pocket, as well as to investigate the manner in which the ligand adapts to its surroundings within the binding pocket.

**RMSD stability study of Indolizine, 2-(4-methylphenyl).**   The stability of the protein backbone [44], protein-ligand complex (com), and the ligand were thoroughly investigated during the simulation. The root-mean-square deviation (RMSD) values range from 0.4 nm to 0.6 nm for protein-ligand complex as depicted in **Fig 2B**. RMSD exhibited a deflection at around 65 nanoseconds for the systems under investigation. This deflection led to a little increase and subsequent drop in the RMSD of Indolizine, 2-(4-methylphenyl). The complex formed between the Indolizine molecule and 2-(4-methylphenyl)-RET was shown to be in a state of equilibrium and exhibited a consistent behavior, characterized by an average value of 0.63 nm. Significantly, RMSD of the complex exhibited increased oscillations from 65ns to 90ns, followed by a subsequent decrease and subsequent stability. The observed discrepancy is hypothesized to arise from the ligand's interaction with the protein's binding pocket. RMSD data indicate that the profiles were adequately equilibrated and have exhibited consistent stability.

**Protein backbone exhibited a compact conformation for 100ns simulation.**   The radius of gyration (Rg) quantifies the degree of compactness shown by the protein, specifically with respect to its backbone structure. Rg for both systems was observed to fall within the range of 2.0 nm to 2.09 nm, respectively. This measurement provides insight into the protein's level of compactness during the simulation as depicted in **Fig 5A**. The calculated average radius of gyration (Rg) for Indolizine, 2-(4-methylphenyl) was determined to be 0.344 nm as illustrated in **Fig 5B**. The measured Rg profile of the reference exhibited an increase from 15ns to 35ns, which was also noted in the RMSD. Subsequently, the system underwent a thorough equilibration process during the simulation. In contrast, it was observed that the protein backbone of Indolizine, namely 2-(4-methylphenyl), exhibited a notably condensed conformation throughout the simulation, as seen by an average distance of 2.07 nm.

**Detection of RMSF with protein backbone.**   The variations of the protein backbone residue atoms were assessed using the Root Mean Square Fluctuation (RMSF) method shown in **Fig 2C**. The residues of the backbone exhibit minimal variations. RMSF values of the residues were estimated to be less than 0.3 nm, indicating a high level of stability in the protein backbone. This finding is consistent with the results obtained from the root mean square deviation (RMSD) and radius of gyration (Rg) analyses. A slight elevation was seen in the residue LEU139 in comparison to the other residues, measuring around 0.49 nm with respect to the backbone and 0.57 nm with respect to the protein. Nevertheless, this residue is not present in

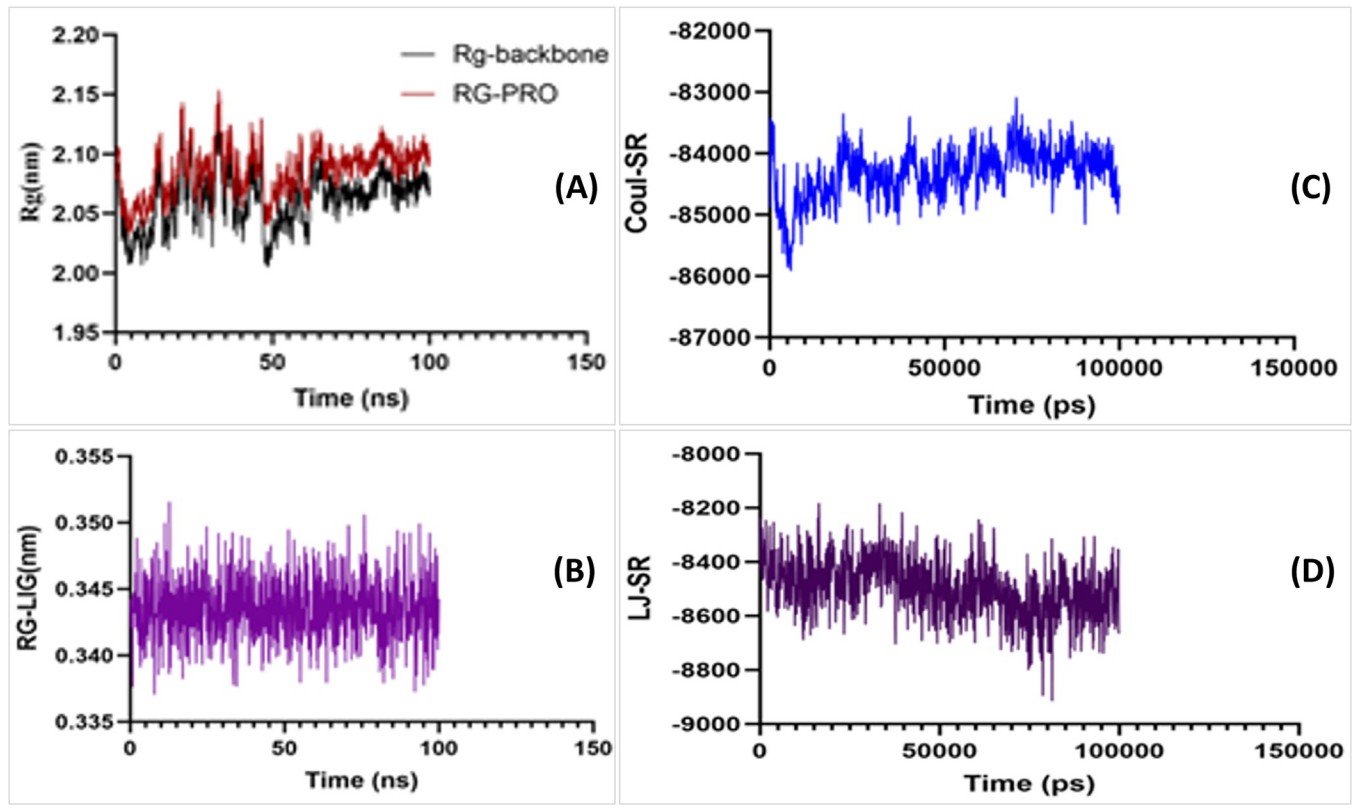

**Fig 5. Radius of gyration (Rg) and protein-ligand interaction energy (IE).** (A) Radius of gyration (Rg) for protein (B) Radius of gyration (Rg) for ligand (C) Coulombic interaction energy (CIE), (D) Lennard-Jones interaction energy (LJIE).

close proximity to the binding pocket. The stability of Indolizine, 2-(4-methylphenyl), was assessed by examining the residues, namely those present in the active site. The distances between these residues were measured and found to be as follows: 0.102 nm for Asp193, 0.0816 nm for Ile189, 0.117 nm for Lys59, 0.0829 nm for Ser192, 0.0905 nm for Val105, 0.0752 nm for Leu182, 0.0672 nm for Gly111, and 0.255 nm for Leu131.

**Detection of better interaction energy for protein-ligand complex.** The term "interaction energy" (IE) refers to the capacity of the ligand to bind to the active site protein. In this study, the IE was determined using two methods: Lennard-Jones interaction energy (LJIE) and Coulombic interaction energy (CIE). The CIE values for Indolizine, 2-(4-methylphenyl), have been seen to vary within the range of -84717.6 kJ/mol to -84887.4 kJ/mol, with an average value of -84322.4 kJ/mol. Additionally, the CIE plots for this compound exhibit a similar pattern to the plots of the calculated lowest LJIE for Indolizine, 2-(4-methylphenyl), showing a decrease in energy after 57000ps depicted in **Fig 5C**. The lower limit of the LJIE for Indolizine, namely 2-(4-methylphenyl), has been seen to range from -8236.42 KJ/mol to -8662.04 KJ/mol, with an average value of -8493.2 KJ/mol illustrated in **Fig 5D**. Based on the observed interaction energies, it is justifiable to infer that the ligand exhibits interaction with the protein.

**MMPBSA analysis.** The MMPBSA analysis showed the calculation of free binding energy ($G_{binding}$). The results depicted that the total binding free energy calculation involving the residues Asp193, Ile189, Lys59, Ser192, Val105, Leu182, Gly111, and Leu131 yields a value of -12.20 kcal/mol. This suggests a significant interaction between the protein RET and Indolizine, 2-(4-methylphenyl). Notably, the van der Waals interaction energy is found to be the

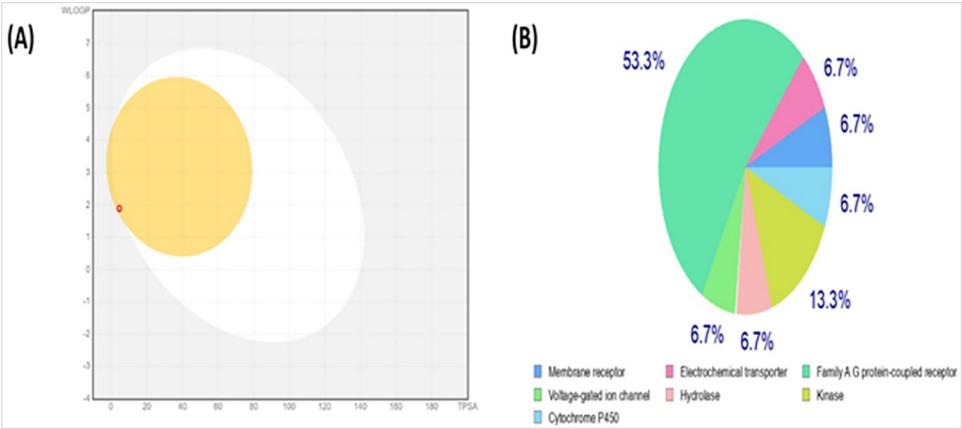

**Fig 6. SwissADME analysis.** (A) Boiled egg analysis of Indolizine, 2-(4-methylphenyl)- (B) Swiss Target Prediction.

highest, with a value of -25.35 kcal/mol, followed by the electrostatic energy with a value of -9.11 kcal/mol. These findings highlight the strong interaction between the key protein residues and the ligand.

**Preclinical investigation.** SwissADME analysis showed that Indolizine, 2-(4-methylphenyl)- is the best anti-cancer compound against NSCLC as it has good lipophilicity Log p less than 4 Log $P_{o/w}$ (2.90). The pharmacokinetic interpretation revealed good GI absorption, water solubility and abide by all the rules of Lipinski's and does not violate it. According to pharmacokinetic analysis and compound interaction showed low glycoprotein permeability and CYP2C9 and CYP2C19 had no inhibitory effect. A BIOLED-EGG image of Indolizine, 2-(4-methylphenyl)- depicted in **Fig 6A** and Swiss Target Prediction are shown in **Fig 6B**.

**Toxicity screening.** The results depicted that there is low predictability against hepatoxicity, carcinogenicity, and immunotoxicity. The mutagenicity and cytotoxicity score are also very low and inactive. **Table 8** below illustrate that there is a high indication that Indolizine, 2-(4-methylphenyl)- will not be carcinogenic, mutagenic, or hepatotoxic.

## Discussion

Genomics and metabolomics have become powerful tools in the discovery of new drugs, particularly in identifying peptide-based drugs derived from natural sources like *C. militaris*. Metabolomics complements this by profiling the metabolites produced, offering insights into the biochemical pathways and potential therapeutic compounds [45,46]. Together, these approaches have facilitated the identification of novel peptide-based drugs from *C. militaris* with promising therapeutic effects, such as antimicrobial, anticancer, and immunomodulatory activities [47,48]. This integrated strategy enhances the discovery process, enabling the development of targeted and effective peptide-based therapies derived from this medicinal fungus [49].

*Cordyceps militaris* have many potent compounds that are already reported [50] but there is no work available on biometabolites analysis for methanol extract. So, in this study we investigated the biometabolites of methanol extract and antimicrobial peptides from medicinal mushroom to eliminate the risks of microbial infections and other ailments. These biometabolites and antimicrobial peptides have potential of biological activities. Analysis of FTIR spectroscopy of current studies of methanolic extract of *C. militaris* was evaluated and different bioactive components were confirmed by FTIR spectroscopy, including phenols, carboxylic acid, amines and alkanes which supports the biologically active compounds identified through

**Table 8. Toxicity analysis of compound Indolizine, 2-(4-methylphenyl)- from Protox-II.** Prediction status: Inactive.

| Toxicity Model Report of Drug | | | |
|---|---|---|---|
| Classification | Target | Shorthand | Probability |
| Organ toxicity | Hepatotoxicity | Dili | 0.84 |
| Toxicity end points | Carcinogenicity | Carcino | 0.74 |
| | Immunotoxicity | Immune | 0.78 |
| | Mutagenicity | Mutagen | 0.67 |
| | Cytotoxicity | cyto | 0.60 |
| Tox21-Nuclear receptor signalling pathways | Androgen Receptor (AR) | nr_ar | 0.97 |
| | Aryl hydrocarbon Receptor | nr_ahr | 0.90 |
| | Aromase | nr_aromase | 0.96 |
| | Androgen Receptor Ligand Binding Domain (AR- LBD) | nr_ar_lbd | 0.99 |
| | Estrogen Receptor Ligand Binding Domain (ER-LBD) | nr_er_lbd | 0.97 |
| | Estrogen Receptor Alpha (ER) | nr_er | 0.92 |
| | Nuclear factor (erythroid-derived 2)-like 2/antioxidant responsive element (nrf2/ARE) | sr_are | 0.98 |
| | Peroxisome Proliferator Activated Receptor Gamma (PPAR-Gamma) | nr_ppar_gamma | 0.98 |
| | Mitochondrial Membrane Potential (MMP) | sr_mmp | 0.93 |
| | Heat shock factor response element | sr_hse | 0.98 |
| | ATPase family AAA domain-containing protein 5 (ATAD5) | sr_atad5 | 0.98 |
| | Phosphoprotein (Tumor Suppressor) p53 | sr_p53 | 0.95 |

The green color represents inactive toxicity based on a likelihood value from 0 to 10. A higher number indicates less toxicity.

GC-MS and LC-MS. The powerful instance peak with the O-H stretching at 3323 cm$^{-1}$ were observed. Previous study showed two strong absorption peaks at 3397 cm$^{-1}$ and 3192 cm$^{-1}$ with the O-H stretching vibrations [51]. These peaks are characteristics absorption peaks for polysaccharides.

GC-MS analysis shows the chemical composition of methanolic extract of *C. militaris*. Thirty-four compounds show 100% of total detected constituents were identified. The major compounds were 1,2,3-Benzenetriol (21.77%), 5-Hydroxymethylfurfural (19.61%), Methyl 3,4,5-trihydroxybenzoate (5.45%), 2(5H)-Furanone (3.70%), and 2-Furancarboxylic acid (2.65%). The chemical composition determined by us is different from other studies. Few studies conducted in Italy, Poland, Spain and Iran, respectively, reported the major components of *C. militaris* were 4,5-Diamino-6-hydroxypyrimidine (29.01), Cyclobutanol (14.5), Trans-Cinnamic acid (30.78), and 1,6-Anhydro-beta-D-glucopyranose (40.1) [52]. In our study LC-MS analysis showed various bioactive compounds in methanol extract of *C. militaris* as malonic acid, 1,2,3-Trihydroxybenzene, 5-Ethoxy-4,5-dihydro-2(3H)-furanone, Caftaric acid, 4-Ethoxy-4-oxobutanoic acid, Oxoadipic acid, Trans-Aconitic acid, 3-Methyl-4-hydroxy-5-all-trans-heptaprenylbenzoate and Ampeloside Bs1 by comparison to retention time. According to LC-MS analysis reported in previous studies showed the seven phenolic acids (chlorogenic acid, protocatechuic acid, gallic acid, coumaric acid, and hydroxybenzoic acid) and four flavonoids (quercetin, naringenin, apigenin, and rutin) reported in methanol extract of *C. militaris* [53].

Research findings show the antioxidant potential of *C. militaris*. The antioxidant capacity of *C. militaris* was measured by using DPPH assay which in non-enzymatic assay. The scavenging activity was increased with the increased of time. The antioxidant activity of *C. militaris* of methanolic extract fractions at different concentrations (12, 6, 3, 1.5, 0.75 mg/mL) were assessed by DPPH radical scavenging assay, evaluating the hydrogen transfer or free radical scavenging ability of the methanolic extracts using DPPH reagent and we compared their

activity to the standard ascorbic acid The result indicated that methanolic extract of *C. militaris* show a high percentage of inhibition (80 ± 0.03) at 12 mg/ml concentration and lowest (55 ± 0.09) at 0.75 mg/ml concentration. The free radical scavenging ability of *C. militaris* varies remarkably ($p < 0.005$). These results are consistent with the results of a previous study [54].

In *Escherichia coli*, methanol extract manifests the maximum zone of inhibition (6.9 ± 0.04). While (6.5 ± 0.01), (3 ± 0.98) and (3.2 ± 0.42) zone of inhibition against bacterial strains *Klebsiella pneumoniae*, *Streptococcus viridans*, and *Staphylococcus aureus* respectively. While the protein Fractions of 70%, 50% and 30% showed visible zone of inhibition i.e., (4 ± 0.22), (3.5 ± 0.06) and (3.6 ± 0.01) in *Escherichia coli* respectively. In *Klebsiella pneumonia*, 70%, 50%, and 30% fraction showed the (2.7 ± 0.06), (0.5 ± 0.09) and (0.2 ± 0.03) zone of inhibition respectively. The highest antibacterial activity found against *Staphylococcus aureus* which showed the maximum zone of inhibition in 70% i.e., (5.4 ± 0.08), while 50% and 30% was (3 ± 0.02) and (1 ± 0.04) respectively. In *Streptococcus viridans*, 70%, 50% and 30% fraction showed the (4.3 ± 0.034), (3.3 ± 0.07) and (4 ± 0.176) zone of inhibition respectively. The ability of inhibition of bacterial strains of fractions of *C. militaris* varies significantly ($p < 0.05$).

Current study showed the antidiabetic activity of methanolic extract of *C. militaris* at different concentrations (12, 6, 3, 1.5, 0.75 mg/mL). The highest antidiabetic activity was shown (37 ± 0.057) at 12 mg/mL and lowest activity (13 ± 0.084) at 0.75 mg/ml. The percentage inhibition of antidiabetic activity of *C. militaris* is significantly $p < 0.05$. The protein denaturation is a source to find inflammation. Therefore, to assess the anti-inflammatory mechanism of *C. militaris*, its extract capability to inhibits bovine serum denaturation is calculated at different concentration (12, 6, 3, 1.5, 0.75 mg/mL). It was observed that methanolic extract shows maximum inhibition of denaturation of BSA (40 ± 0.021) which is less than standard diclofenac (86 ± 0.096) at 12 μl/ml concentration. The result shows a concentration-dependent manner on inhibition of protein denaturation by *C. militaris*. The ability to protein inhibition (BSA) of *C. militaris* varies significantly ($p < 0.05$).

Protein extractions were quantified by Bradford assay. Result revealed that extract of 70% concentration contains higher protein concentration (500 μg/ml ± 0.025) than the concentrations of 30 and 50%. The 30% concentration contains lower amount of protein (250 μg/ml ± 0.07). while 50% concentration shows the (450 μg/ml ± 0.012). Protein quantity analysis is important for quality control. The method discovered by the Marion Bradford was based on the absorption of max light by the protein sample bound to the brilliant blue G-250 [55]. The sample with 70% shows the maximum bands compared to the 50 and 30% ranges from ~100 to ~10 kDa. 70% showed cluster of bands with size of 70 kDa, 64 kDa, 59 kDa, 45 kDa, 32 kDa, 22 kDa. 50% showed weak bands with the size of 40 kDa, and 22 kDa. while 30% showed no band. Therefore, it may be inferred from both the prior and present research that *C. militaris* fruiting bodies include dynamic proteins. The different concentration of *C. militaris* extracts treated with digestive enzymes pepsin, trypsin and papain in order to produce the peptides. Enzymatic treatment was used for *C. militaris* isolates to form peptides. Pepsin (37˚C and pH 2–4), Trypsin (65˚C and pH 9) and Papain (37–80˚C and pH 7) enzyme was used to make the peptides of most abundant protein in the *C. militaris*. SDS-PAGE analysis of the enzyme treated extract showed only trypsin digestion and show peptides in lower kDa.

The results of the molecular docking studies indicate that among the several compounds examined, 2-(4-methylphenyl) Indolizine exhibited a superior dock score, accompanied by the highest cluster. The co-crystallized structure obtained from the X-ray structure is utilized for conducting a molecular dynamics (MD) simulation investigation. The dock score (-CDOCKER interaction energy, kcal/mol) of Indolizine, 2-(4-methylphenyl) was -8.0 kcal/mol with RET gene. The pose with the highest score within this cluster was selected as the

optimal pose and subsequently subjected to Molecular Dynamics Simulations (MDS) in order to evaluate its stability for the whole duration of the simulation run. MD simulation was performed by using GROMACS.

The RMSD stability study of Indolizine, 2-(4-methylphenyl) showed the values of the protein-ligand complex range from 0.4 nm to 0.6 nm. The complex formed between the Indolizine molecule and 2-(4-methylphenyl)-RET was shown to be in a state of equilibrium and exhibited a consistent behavior, characterized by an average value of 0.63 nm. Significantly, RMSD of the complex exhibited increased oscillations from 65ns to 90ns, followed by a subsequent decrease and subsequent stability. Radius of gyration ($R_g$) quantifies the degree of compactness shown by the protein, specifically with respect to its backbone structure. $R_g$ for both systems was observed to fall within the range of 2.0 to 2.09 nm, respectively. The calculated average radius of gyration (Rg) for Indolizine, 2-(4-methylphenyl) was determined to be 0.344 nm. The measured Rg profile of the reference exhibited an increase from 15ns to 35ns, which was also noted in the RMSD. Subsequently, the system underwent a thorough equilibration process during the simulation. In contrast, it was observed that the protein backbone of Indolizine, namely 2-(4-methylphenyl), exhibited a notably condensed conformation throughout the simulation, as seen by an average distance of 2.07 nm.

The variations of the protein backbone residue atoms were assessed using the RMSF method. RMSF values of the residues were estimated to be less than 0.3 nm, indicating a high level of stability in the protein backbone. This finding is consistent with the results obtained from the RMSD and Rg analyses. A slight elevation was seen in the residue LEU139 in comparison to the other residues, measuring around 0.49 nm with respect to the backbone and 0.57 nm with respect to the protein. Nevertheless, this residue is not present in close proximity to the binding pocket. The stability of Indolizine, 2-(4-methylphenyl), was assessed by examining the residues, namely those present in the active site. The distances between these residues were measured and found to be as follows: 0.102nm for Asp193, 0.0816nm for Ile189, 0.117nm for Lys59, 0.0829nm for Ser192, 0.0905nm for Val105, 0.0752nm for Leu182, 0.0672nm for Gly111, and 0.255nm for Leu131.

The CIE values for Indolizine, 2-(4-methylphenyl), have been seen to vary within the range of -84717.6 kJ/mol to -84887.4 kJ/mol, with an average value of -84322.4 kJ/mol. Additionally, the CIE plots for this compound exhibit a similar pattern to the plots of the calculated lowest LJIE for Indolizine, 2-(4-methylphenyl), showing a decrease in energy after 57000 ps. The lower limit of the LJIE for Indolizine, namely 2-(4-methylphenyl), has been seen to range from -8236.42 KJ/mol to -8662.04 KJ/mol, with an average value of -8493.2 KJ/mol. Based on the observed interaction energies, it is justifiable to infer that the ligand exhibits interaction with the protein.

The total binding free energies of the selected bioactive molecule were determined using the MMPBSA method which showed that RET–Indolizine, 2-(4-methylphenyl) complex depicts the total binding energy (BFE), Van der Waals energy, electrostatic energy, polar solvation energy contains -12.20, -25.35, -9.11 and -2.66 respectively. The calculation of free binding energy ($G_{binding}$) results indicates that the total binding free energy calculation involving the residues Asp193, Ile189, Lys59, Ser192, Val105, Leu182, Gly111, and Leu131 yields a value of -12.20 kcal/mol. This suggests a significant interaction between the protein RET and Indolizine, 2-(4-methylphenyl). Notably, the van der Waals interaction energy is found to be the highest, with a value of -25.35 kcal/mol, followed by the electrostatic energy with a value of -9.11 kcal/mol. These findings highlighted the strong interaction between the key protein residues and the ligand.

ADMET analysis shows strong drug-likeness and pharmacokinetic properties. It followed all five rule of Lipinski and no violation. According to pharmacokinetic analysis and

compound interaction, it showed low glycoprotein permeability and CYP2C9 and CYP2C19 had no inhibitory effect. Furthermore, toxicity screening was performed through Protox-II tool and it depicted that there is low predictability against hepatoxicity, carcinogenicity, and immunotoxicity of Indolizine, 2-(4-methylphenyl)-. The mutagenicity and cytotoxicity score are also very low and inactive. Table 8 illustrate that there is a high indication that Indolizine, 2-(4-methylphenyl)- will not be carcinogenic, mutagenic, or hepatotoxic.

## Conclusion

This study was used to analyze the bioactivity of *C. militaris*. The test includes the antioxidant activity using non-enzymatic assay, antimicrobial activity using resistant and non-resistant strains. In biological activities, methanol extract of *C. militaris* showed strong antioxidant activity as cordycepin and other phenolic compounds attributed many of the antioxidant properties to the compounds due to their strong hydrogen donating ability. The extract showed a little activity because inhibition zone diameter was too small. Therefore, these findings suggest that *C. militaris* is an excellent candidate for the treatment of various ailments especially those for which antioxidant stress is one of the mechanisms for its occurrence. It was revealed that the indolizine, 2-(4-methylphenyl) has most binding affinity (micromolar) and optimized properties to be selected as the lead inhibitor. It interacts favorably with the active site of RET gene of NSCLC and is neuroprotective and hepatoprotective. So, the Indolizine, 2-(4-methylphenyl) is the potential inhibitor of RET gene to treat the N NSCLC. So, it was concluded that the pharmacological and medicinal potentials of *C. militaris* are auspicious as a versatile therapeutic mushroom due to the presence of bioactive compounds and it should be further investigated.

## Supporting information

**S1 Fig. Chromatogram of GC-MS analysis.**
(JPG)

**S2 Fig. Antibacterial activity of methanolic extract of *C. militaris* against bacterial strains.**
(A) *E. coli* (B) *K. pneumoniae* (C) *S. viridans* (D) *S. aureus*.
(PNG)

**S1 Table. GC-MS analysis of methanol extract of *C. militaris*.**
(PDF)

**S2 Table. LC-MS analysis of methanol extract of *C. militaris*.**
(PDF)

**S3 Table. Binding pockets position and their scores of Aldose Reductase protein by Deep-Site.**
(PDF)

**S4 Table. Molecular docking of bioactive compound of GC-MS analysis with RET, PIK3CA and TCTN3 gene.**
(PDF)

## Acknowledgments

The authors greatly acknowledge and express their gratitude to the Research work which was supported by University of Central Punjab, Lahore, Pakistan and collaboration with The First Affiliated Hospital of Zhengzhou University & Institute of Reproductive Health, Henan

Academy of Innovations in Medical Science, NHC Key Laboratory of Birth Defects Prevention & Institute of drug discovery and development, Zhengzhou University, Zhengzhou 450001, China in computationally analysis.

## Author Contributions

**Conceptualization:** Muhammad Afzal, Kirnpal Kaur Banga Singh.

**Data curation:** Mai Abdel Haleem A. Abusalah, Muhammad Absar, Naveed Ahmed, Noshaba Mehmood, Kirnpal Kaur Banga Singh.

**Formal analysis:** Neelum Shehzadi, Muhammad Absar, Sarmir Khan, Noshaba Mehmood.

**Investigation:** Neelum Shehzadi, Noshaba Mehmood.

**Methodology:** Neelum Shehzadi, Noshaba Mehmood.

**Project administration:** Muhammad Afzal.

**Resources:** Mai Abdel Haleem A. Abusalah, Naveed Ahmed, Yalnaz Naseem, Kirnpal Kaur Banga Singh.

**Software:** Mai Abdel Haleem A. Abusalah, Sarmir Khan, Yalnaz Naseem.

**Supervision:** Muhammad Afzal.

**Validation:** Muhammad Afzal, Mai Abdel Haleem A. Abusalah, Muhammad Absar, Naveed Ahmed, Kirnpal Kaur Banga Singh.

**Visualization:** Naveed Ahmed, Sarmir Khan.

**Writing – original draft:** Neelum Shehzadi, Sarmir Khan, Yalnaz Naseem, Noshaba Mehmood.

**Writing – review & editing:** Muhammad Afzal, Mai Abdel Haleem A. Abusalah, Muhammad Absar, Naveed Ahmed, Kirnpal Kaur Banga Singh.

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
