## [Decision Letter · Decision Letter 0]

18 Mar 2024

PONE-D-24-02007Investigation of Biometabolites and Novel Antimicrobial Peptides Derived from Promising Source Cordyceps militaris and Effect of Non-Small Cell Lung Cancer Genes ComputationallyPLOS ONE

Dear Dr. Ahmed,

Thank you for submitting your manuscript to PLOS ONE. After careful consideration, we feel that it has merit but does not fully meet PLOS ONE’s publication criteria as it currently stands. Therefore, we invite you to submit a revised version of the manuscript that addresses the points raised during the review process.

We look forward to receiving your revised manuscript.

Kind regards,

Jameel Al-Obaidi

Academic Editor

PLOS ONE

Journal Requirements:

3. Please include a copy of Table 12 and 13 which you refer to in your text on page 36 and 37 on PDF submission.

**Additional Editor Comments:**

**ACADEMIC EDITOR:**

Dear Author

I am writing to inform you about the decision regarding your manuscript titled "Investigation of Biometabolites and Novel Antimicrobial Peptides Derived from Promising Source Cordyceps militaris and Effect of Non-Small Cell Lung Cancer Genes Computationally," which has been under review.

After a thorough evaluation and consideration of the respected reviewers' comments, I am pleased to inform you that can be accepted with minor corrections. We believe that your research provides significant contributions to the field, especially in understanding the potential of Cordyceps militaris against Non-Small Cell Lung Cancer genes.

The reviewers have provided feedback aimed at enhancing the clarity, accuracy, and impact of your manuscript. To proceed, we kindly request that you address the following minor corrections suggested by the reviewers (Attached)

Reviewers' comments:

Reviewer's Responses to Questions

**Comments to the Author**

1. Is the manuscript technically sound, and do the data support the conclusions?

Reviewer #1: Yes

Reviewer #2: Yes

2. Has the statistical analysis been performed appropriately and rigorously? 

Reviewer #1: Yes

Reviewer #2: Yes

3. Have the authors made all data underlying the findings in their manuscript fully available?

Reviewer #1: Yes

Reviewer #2: Yes

4. Is the manuscript presented in an intelligible fashion and written in standard English?

Reviewer #1: Yes

Reviewer #2: Yes

5. Review Comments to the Author

Reviewer #1: The manuscript explores a novel and promising research direction by investigating biometabolites and antimicrobial peptides derived from Cordyceps militaris, a source with significant potential for therapeutic applications. The integration of computational analysis to explore the effects on non-small cell lung cancer genes adds depth and relevance to the study, contributing to the advancement of both biotechnology and biomedical research fields.

The paper demonstrates a comprehensive experimental approach to isolate, characterize, and evaluate biometabolites and antimicrobial peptides from Cordyceps militaris. The detailed methodologies provided offer transparency and reproducibility, laying a solid foundation for future investigations in this area. The computational analysis of the interactions between biometabolites/antimicrobial peptides and non-small cell lung cancer genes is a notable strength of the study. By employing computational modeling and bioinformatics tools, the paper expands our understanding of potential therapeutic mechanisms and provides insights into the molecular pathways involved. The results section is well-structured and effectively presents the findings of the study. The inclusion of tables, figures, and supplementary data enhances the clarity and visualization of the experimental outcomes, facilitating interpretation and comparison with existing literature.

The discussion section offers insightful interpretations of the experimental results and effectively contextualizes them within the broader scientific literature. The authors critically analyze the implications of their findings and propose intriguing avenues for future research, stimulating further inquiry and innovation in the field.

In summary, the manuscript represents a significant advancement in our understanding of Cordyceps militaris-derived biometabolites and antimicrobial peptides, as well as their potential impact on non-small cell lung cancer. The combination of experimental and computational approaches, coupled with clear presentation and thoughtful interpretation of results, positions the study as a valuable contribution to both basic and translational research efforts in biotechnology and cancer biology.

Reviewer #2: Dear author please refer to the comments in the manuscript attached.

Dear Author,

Good day,

I am hereby writing my feedback on the scientific paper entitled “Investigation of Biometabolites and Novel Antimicrobial Peptides Derived from Promising Source Cordyceps militaris and Effect of Non-Small Cell Lung Cancer Genes Computationally “.

The manuscript is of good value due to its important character, it treats a specific subject that is of high interest for public health.

The present manuscript approaches an interesting topic, that is, to analyze the potentials of biometabolites and to extract antimicrobial peptides and protein from the C. militaris..

Thereby I would like to inform you that the paper is qualified to publish in this scientific journal with some minor revision as shown in the comments in the attached manuscripts .

6. PLOS authors have the option to publish the peer review history of their article (what does this mean?). If published, this will include your full peer review and any attached files.

Reviewer #1: **Yes: **Assistant Professor Dr. Rana I. Mahmood

Reviewer #2: No

---

## [Author Response · Author response to Decision Letter 0]

26 Mar 2024

ACADEMIC EDITOR:

Dear Author

I am writing to inform you about the decision regarding your manuscript titled "Investigation of Biometabolites and Novel Antimicrobial Peptides Derived from Promising Source Cordyceps militaris and Effect of Non-Small Cell Lung Cancer Genes Computationally," which has been under review.

After a thorough evaluation and consideration of the respected reviewers' comments, I am pleased to inform you that can be accepted with minor corrections. We believe that your research provides significant contributions to the field, especially in understanding the potential of Cordyceps militaris against Non-Small Cell Lung Cancer genes.

Response: Dear editor, we would like to thank you for your kind efforts to review our manuscript and recommend it towards the possible publication process. We hope that this work will contribute to the understanding of scientific community and will benefit the reader.

The reviewers have provided feedback aimed at enhancing the clarity, accuracy, and impact of your manuscript. To proceed, we kindly request that you address the following minor corrections suggested by the reviewers (Attached)

3. Please include a copy of Table 12 and 13 which you refer to in your text on page 36 and 37 on PDF submission.

Response: Dear editor, thank you for highlighting this mistake in our manuscript. We have correctly cited the table number in the revised version of manuscript. For table 12, we have removed it and defined the information as text.

Response: Dear editor, we have checked back all of the references and corrected wherever needs any corrections. 

Reviewer #1: The manuscript explores a novel and promising research direction by investigating biometabolites and antimicrobial peptides derived from Cordyceps militaris, a source with significant potential for therapeutic applications. The integration of computational analysis to explore the effects on non-small cell lung cancer genes adds depth and relevance to the study, contributing to the advancement of both biotechnology and biomedical research fields.

The paper demonstrates a comprehensive experimental approach to isolate, characterize, and evaluate biometabolites and antimicrobial peptides from Cordyceps militaris. The detailed methodologies provided offer transparency and reproducibility, laying a solid foundation for future investigations in this area. The computational analysis of the interactions between biometabolites/antimicrobial peptides and non-small cell lung cancer genes is a notable strength of the study. By employing computational modeling and bioinformatics tools, the paper expands our understanding of potential therapeutic mechanisms and provides insights into the molecular pathways involved. The results section is well-structured and effectively presents the findings of the study. The inclusion of tables, figures, and supplementary data enhances the clarity and visualization of the experimental outcomes, facilitating interpretation and comparison with existing literature. The discussion section offers insightful interpretations of the experimental results and effectively contextualizes them within the broader scientific literature. The authors critically analyze the implications of their findings and propose intriguing avenues for future research, stimulating further inquiry and innovation in the field.

In summary, the manuscript represents a significant advancement in our understanding of Cordyceps militaris-derived biometabolites and antimicrobial peptides, as well as their potential impact on non-small cell lung cancer. The combination of experimental and computational approaches, coupled with clear presentation and thoughtful interpretation of results, positions the study as a valuable contribution to both basic and translational research efforts in biotechnology and cancer biology.

Response: Dear reviewer, we would like to thank you for your kind efforts to review our manuscript and recommend it towards the possible publication process. We hope that this work will contribute to the understanding of scientific community and will benefit the reader.

Reviewer #2: Dear author please refer to the comments in the manuscript attached.

Dear Author,

Good day,

I am hereby writing my feedback on the scientific paper entitled “Investigation of Biometabolites and Novel Antimicrobial Peptides Derived from Promising Source Cordyceps militaris and Effect of Non-Small Cell Lung Cancer Genes Computationally “. The manuscript is of good value due to its important character, it treats a specific subject that is of high interest for public health. The present manuscript approaches an interesting topic, that is, to analyze the potentials of biometabolites and to extract antimicrobial peptides and protein from the C. militaris. Thereby I would like to inform you that the paper is qualified to publish in this scientific journal with some minor revision as shown in the comments in the attached manuscripts.

Response: Dear reviewer, we would like to thank you for your kind efforts to review our manuscript and recommend it towards the possible publication process. We hope that this work will contribute to the understanding of scientific community and will benefit the reader. Your comments have been addressed in the revised version of manuscript.

---

## [Decision Letter · Decision Letter 1]

13 Aug 2024

PONE-D-24-02007R1Investigation of Biometabolites and Novel Antimicrobial Peptides Derived from Promising Source Cordyceps militaris and Effect of Non-Small Cell Lung Cancer Genes ComputationallyPLOS ONE

Dear Dr. Ahmed,

Thank you for submitting your manuscript to PLOS ONE. After careful consideration, we feel that it has merit but does not fully meet PLOS ONE’s publication criteria as it currently stands. Therefore, we invite you to submit a revised version of the manuscript that addresses the points raised during the review process.

We look forward to receiving your revised manuscript.

Kind regards,

Afzal Basha Shaik, Ph.D

Academic Editor

PLOS ONE

Journal Requirements:

**Additional Editor Comments:**

The authors have revised the manuscript according to the comments provided by the reviewers. However, it still needs minor revisions as per the additional reviewers comments and my concerns. I recommend to enrich the introduction part by including the following contents and references.

1. Genomics and metabolomics have been extensively investigated for the discovery of new drugs particularly in the discovery of peptide-based drugs. Add few lines about the same and cite the below references

a. Jiang, M., Chen, S., Lu, X., Guo, H., Chen, S., Yin, X.,... Liu, L. (2023). Integrating Genomics and Metabolomics for the Targeted Discovery of New Cyclopeptides with Antifungal Activity from a Marine-Derived Fungus Beauveria felina. Journal of Agricultural and Food Chemistry, 71(25), 9782-9795. doi: 10.1021/acs.jafc.3c02415.

b. Kloosterman, A.M., Medema, M.H. and van Wezel, G.P., 2021. Omics-based strategies to discover novel classes of RiPP natural products. Current Opinion in Biotechnology, 69, pp.60-67.

c. Xu, T., Yan, X., Kang, A., Yang, L., Li, X., Tian, Y.,... Guo, Y. (2024). Development of Membrane-Targeting Fluorescent 2-Phenyl-1H-phenanthro[9,10-d]imidazole-Antimicrobial Peptide Mimic Conjugates against Methicillin-Resistant Staphylococcus aureus. Journal of Medicinal Chemistry, 67(11), 9302-9317. doi: https://doi.org/10.1021/acs.jmedchem.4c00436

d. He, B., Lang, J., Wang, B., Liu, X., Lu, Q., He, J.,... Yang, J. (2020). TOOme: A Novel Computational Framework to Infer Cancer Tissue-of-Origin by Integrating Both Gene Mutation and Expression. Frontiers in Bioengineering and Biotechnology, 8. doi: 10.3389/fbioe.2020.00394

e. Ali, A.M., Atmaj, J., Van Oosterwijk, N., Groves, M.R. and Dömling, A., 2019. Stapled peptides inhibitors: a new window for target drug discovery. Computational and structural biotechnology journal, 17, pp.263-281.

2. Some of the important works related to the therapeutic modalities in non-small cell lung cancer were not referred. These must be discussed and cited in the introduction.

a. Zhao, Y., Chen, S., Shen, F., Long, D., Yu, T., Wu, M.,... Lin, X. (2019). In vitro neutralization of autocrine IL‑10 affects Op18/stathmin signaling in non‑small cell lung cancer cells. Oncol Rep, 41(1), 501-511. doi: 10.3892/or.2018.6795

b. Chen, S., Zhao, Y., Shen, F., Long, D., Yu, T.,... Lin, X. (2019). Introduction of exogenous wild‑type p53 mediates the regulation of oncoprotein 18/stathmin signaling via nuclear factor‑κB in non‑small cell lung cancer NCI‑H1299 cells. Oncol Rep, 41(3), 2051-2059. doi: 10.3892/or.2019.6964

c. Dai, J., Ashrafizadeh, M., Aref, A. R., Sethi, G., & Ertas, Y. N. (2024). Peptide-functionalized, -assembled and -loaded nanoparticles in cancer therapy. Drug Discovery Today, 29(7), 103981. doi: https://doi.org/10.1016/j.drudis.2024.103981

d. Tan, T., Feng, Y., Wang, W., Wang, R., Yin, L., Zeng, Y.,... Xie, T. (2023). Cabazitaxel-loaded human serum albumin nanoparticles combined with TGFβ-1 siRNA lipid nanoparticles for the treatment of paclitaxel-resistant non-small cell lung cancer. Cancer Nanotechnology, 14(1), 70. doi: https://doi.org/10.1186/s12645-023-00194-7

e. Lou, J., Zhao, L., Huang, Z., Chen, X., Xu, J., TAI, W. C.,... Xie, T. (2021). Ginkgetin derived from Ginkgo biloba leaves enhances the therapeutic effect of cisplatin via ferroptosis-mediated disruption of the Nrf2/HO-1 axis in EGFR wild-type non-small-cell lung cancer. Phytomedicine, 80, 153370. doi: https://doi.org/10.1016/j.phymed.2020.153370

Reviewers' comments:

Reviewer's Responses to Questions

**Comments to the Author**

1. If the authors have adequately addressed your comments raised in a previous round of review and you feel that this manuscript is now acceptable for publication, you may indicate that here to bypass the “Comments to the Author” section, enter your conflict of interest statement in the “Confidential to Editor” section, and submit your "Accept" recommendation.

Reviewer #1: All comments have been addressed

Reviewer #2: All comments have been addressed

Reviewer #3: (No Response)

Reviewer #4: (No Response)

2. Is the manuscript technically sound, and do the data support the conclusions?

Reviewer #1: Yes

Reviewer #2: Yes

Reviewer #3: Yes

Reviewer #4: Yes

3. Has the statistical analysis been performed appropriately and rigorously? 

Reviewer #1: Yes

Reviewer #2: Yes

Reviewer #3: Yes

Reviewer #4: Yes

4. Have the authors made all data underlying the findings in their manuscript fully available?

Reviewer #1: Yes

Reviewer #2: Yes

Reviewer #3: Yes

Reviewer #4: Yes

5. Is the manuscript presented in an intelligible fashion and written in standard English?

Reviewer #1: Yes

Reviewer #2: Yes

Reviewer #3: Yes

Reviewer #4: Yes

6. Review Comments to the Author

Reviewer #1: (No Response)

Reviewer #2: (No Response)

Reviewer #3: The topic to which the manuscript “ Investigation of Biometabolites and Novel Antimicrobial Peptides Derived from Promising Source Cordyceps militaris and Effect of Non-Small Cell Lung Cancer Genes Computationally” is devoted is relevant and of practical importance. Due to a wide range of bioactive compounds, C. militaris has significant potential for pharmacological use. The manuscript is of a modern scientific level, impresses with its instrumental level of execution and depth of research.This work definitely deserves to be published and the results highlighted.

However, there are a number of recommendations that would improve the introduction.

The article is devoted to the study of the impact of Cordyceps militaris of Non-Small Cell Lung Cancer Genes. However, the introduction does not contain any information about the already published studies of the effect of this mushroom on non-small cell lung cancer. This information should be added and I recommend pay attention to the following works:

Luo L, Ran R, Yao J, Zhang F, Xing M, Jin M, Wang L, Zhang T. Se-Enriched Cordyceps militaris Inhibits Cell Proliferation, Induces Cell Apoptosis, And Causes G2/M Phase Arrest In Human Non-Small Cell Lung Cancer Cells. Onco Targets Ther. 2019;12:8751-8763 https://doi.org/10.2147/OTT.S217017

Jo E, Jang HJ, Shen L, Yang KE, Jang MS, Huh YH, Yoo HS, Park J, Jang IS, Park SJ. Cordyceps militaris Exerts Anticancer Effect on Non-Small Cell Lung Cancer by Inhibiting Hedgehog Signaling via Suppression of TCTN3. Integr Cancer Ther. 2020 Jan-Dec;19:1534735420923756. doi: 10.1177/1534735420923756.

Lu YY, Huang X, Luo ZC, et al. [Mechanism of Cordyceps militaris against non-small cell lung cancer: based on serum metabolomics]. Zhongguo Zhong yao za zhi = Zhongguo Zhongyao Zazhi = China Journal of Chinese Materia Medica. 2022 Sep;47(18):5032-5039. DOI: 10.19540/j.cnki.cjcmm.20220613.702.

In my opinion, the information about the SDS-PAGE method (line 75-79) is unnecessary in the introduction. It is logical to provide this information in the discussion.

Line 112-113: The mushroom was taxonomically classified, identified and 113 confirmed from the University of Central Punjab, Lahore Pakistan.

Was the sample genetically identified? Was the data entered into the genebank? If yes, please provide the sample number.

Reviewer #4: (No Response)

7. PLOS authors have the option to publish the peer review history of their article (what does this mean?). If published, this will include your full peer review and any attached files.

Reviewer #1: No

Reviewer #2: **Yes: **Dhafar N. Al-ugaili

Reviewer #3: No

Reviewer #4: No

---

## [Author Response · Author response to Decision Letter 1]

17 Aug 2024

Journal Requirements:

Author response: Dear editorial members, thank you so much for allowing us to make the manuscript better for the reader and scientific community. We have revised the manuscript as per the comments from the editor and reviewers and we have attached a rebuttal letter in the submission portal. We have highlighted all of the changes in red colour and in track change format. Furthermore, we have counterchecked the cited references and we did not noticed any retraction for them.

Additional Editor Comments:

The authors have revised the manuscript according to the comments provided by the reviewers. However, it still needs minor revisions as per the additional reviewers comments and my concerns. I recommend to enrich the introduction part by including the following contents and references.

Author response: Dear editor, thank you so much for allowing us to make the manuscript better for the reader and scientific community. We have revised the manuscript as per the comments from the editor and reviewers and we have attached a rebuttal letter in the submission portal. We have highlighted all of the changes in red colour and in track change format.

1. Genomics and metabolomics have been extensively investigated for the discovery of new drugs particularly in the discovery of peptide-based drugs. Add few lines about the same and cite the below references

a. Jiang, M., Chen, S., Lu, X., Guo, H., Chen, S., Yin, X.,... Liu, L. (2023). Integrating Genomics and Metabolomics for the Targeted Discovery of New Cyclopeptides with Antifungal Activity from a Marine-Derived Fungus Beauveria felina. Journal of Agricultural and Food Chemistry, 71(25), 9782-9795. doi: 10.1021/acs.jafc.3c02415.

b. Kloosterman, A.M., Medema, M.H. and van Wezel, G.P., 2021. Omics-based strategies to discover novel classes of RiPP natural products. Current Opinion in Biotechnology, 69, pp.60-67.

c. Xu, T., Yan, X., Kang, A., Yang, L., Li, X., Tian, Y.,... Guo, Y. (2024). Development of Membrane-Targeting Fluorescent 2-Phenyl-1H-phenanthro[9,10-d]imidazole-Antimicrobial Peptide Mimic Conjugates against Methicillin-Resistant Staphylococcus aureus. Journal of Medicinal Chemistry, 67(11), 9302-9317. doi: https://doi.org/10.1021/acs.jmedchem.4c00436

d. He, B., Lang, J., Wang, B., Liu, X., Lu, Q., He, J.,... Yang, J. (2020). TOOme: A Novel Computational Framework to Infer Cancer Tissue-of-Origin by Integrating Both Gene Mutation and Expression. Frontiers in Bioengineering and Biotechnology, 8. doi: 10.3389/fbioe.2020.00394

e. Ali, A.M., Atmaj, J., Van Oosterwijk, N., Groves, M.R. and Dömling, A., 2019. Stapled peptides inhibitors: a new window for target drug discovery. Computational and structural biotechnology journal, 17, pp.263-281.

Author response: Dear editor, thank you for your valuable suggestion to improve the writeup by providing some background about the genomics and metabolomics. We have amended the discussion section and have cited the relevant new references to strengthen the statements. (Line 560-568)

2. Some of the important works related to the therapeutic modalities in non-small cell lung cancer were not referred. These must be discussed and cited in the introduction.

a. Zhao, Y., Chen, S., Shen, F., Long, D., Yu, T., Wu, M.,... Lin, X. (2019). In vitro neutralization of autocrine IL 10 affects Op18/stathmin signaling in non small cell lung cancer cells. Oncol Rep, 41(1), 501-511. doi: 10.3892/or.2018.6795

b. Chen, S., Zhao, Y., Shen, F., Long, D., Yu, T.,... Lin, X. (2019). Introduction of exogenous wild type p53 mediates the regulation of oncoprotein 18/stathmin signaling via nuclear factor κB in non small cell lung cancer NCI H1299 cells. Oncol Rep, 41(3), 2051-2059. doi: 10.3892/or.2019.6964

c. Dai, J., Ashrafizadeh, M., Aref, A. R., Sethi, G., & Ertas, Y. N. (2024). Peptide-functionalized, -assembled and -loaded nanoparticles in cancer therapy. Drug Discovery Today, 29(7), 103981. doi: https://doi.org/10.1016/j.drudis.2024.103981

d. Tan, T., Feng, Y., Wang, W., Wang, R., Yin, L., Zeng, Y.,... Xie, T. (2023). Cabazitaxel-loaded human serum albumin nanoparticles combined with TGFβ-1 siRNA lipid nanoparticles for the treatment of paclitaxel-resistant non-small cell lung cancer. Cancer Nanotechnology, 14(1), 70. doi: https://doi.org/10.1186/s12645-023-00194-7

e. Lou, J., Zhao, L., Huang, Z., Chen, X., Xu, J., TAI, W. C.,... Xie, T. (2021). Ginkgetin derived from Ginkgo biloba leaves enhances the therapeutic effect of cisplatin via ferroptosis-mediated disruption of the Nrf2/HO-1 axis in EGFR wild-type non-small-cell lung cancer. Phytomedicine, 80, 153370. doi: https://doi.org/10.1016/j.phymed.2020.153370

Author response: Dear editor, thank you for your valuable suggestion to improve the discussion section. We have amended the discussion section and have cited the relevant new references to strengthen the statements. A new paragraph has been added on the impact of Cordyceps militaris of Non-Small Cell Lung Cancer Genes. (Line 100-110)

Reviewer #3: 

The topic to which the manuscript “Investigation of Biometabolites and Novel Antimicrobial Peptides Derived from Promising Source Cordyceps militaris and Effect of Non-Small Cell Lung Cancer Genes Computationally” is devoted is relevant and of practical importance. Due to a wide range of bioactive compounds, C. militaris has significant potential for pharmacological use. The manuscript is of a modern scientific level, impresses with its instrumental level of execution and depth of research.This work definitely deserves to be published and the results highlighted. However, there are a number of recommendations that would improve the introduction.

Author response: Dear reviewer, thank you so much for valuable words, to review our manuscript, and to provide us with your valuable comments. The manuscript became better for the reader and scientific community. We have revised the manuscript as per the comments from the you and other reviewers and we have attached a rebuttal letter in the submission portal. We have highlighted all of the changes in red colour and in track change format.

The article is devoted to the study of the impact of Cordyceps militaris of Non-Small Cell Lung Cancer Genes. However, the introduction does not contain any information about the already published studies of the effect of this mushroom on non-small cell lung cancer. This information should be added and I recommend pay attention to the following works:

Luo L, Ran R, Yao J, Zhang F, Xing M, Jin M, Wang L, Zhang T. Se-Enriched Cordyceps militaris Inhibits Cell Proliferation, Induces Cell Apoptosis, And Causes G2/M Phase Arrest In Human Non-Small Cell Lung Cancer Cells. Onco Targets Ther. 2019;12:8751-8763 https://doi.org/10.2147/OTT.S217017

Jo E, Jang HJ, Shen L, Yang KE, Jang MS, Huh YH, Yoo HS, Park J, Jang IS, Park SJ. Cordyceps militaris Exerts Anticancer Effect on Non-Small Cell Lung Cancer by Inhibiting Hedgehog Signaling via Suppression of TCTN3. Integr Cancer Ther. 2020 Jan-Dec;19:1534735420923756. doi: 10.1177/1534735420923756.

Lu YY, Huang X, Luo ZC, et al. [Mechanism of Cordyceps militaris against non-small cell lung cancer: based on serum metabolomics]. Zhongguo Zhong yao za zhi = Zhongguo Zhongyao Zazhi = China Journal of Chinese Materia Medica. 2022 Sep;47(18):5032-5039. DOI: 10.19540/j.cnki.cjcmm.20220613.702.

Author response: Dear reviewer, thank you for your valuable suggestion to improve the introduction section. We have amended the introduction section and have cited the relevant new references to strengthen the statements. Line 100-110.

In my opinion, the information about the SDS-PAGE method (line 75-79) is unnecessary in the introduction. It is logical to provide this information in the discussion.

Author response: Dear reviewer, the sentence has been removed from the revised version of manuscript.

Line 112-113: The mushroom was taxonomically classified, identified and 113 confirmed from the University of Central Punjab, Lahore Pakistan.

Was the sample genetically identified? Was the data entered into the genebank? If yes, please provide the sample number.

Author response: Dear reviewer, in the current study, the plant sample was not genetically identified. Hence, we don’t have any data to be deposited on genbank database. The plant sample was identified by a botanist at University of Central Punjab and was given a voucher number for future references. All of the other experiments were conducted only after the confirmation from the botanist.

---

## [Editor Report · Decision Letter 2]

26 Aug 2024

Investigation of Biometabolites and Novel Antimicrobial Peptides Derived from Promising Source Cordyceps militaris and Effect of Non-Small Cell Lung Cancer Genes Computationally

PONE-D-24-02007R2

Dear Dr. Ahmed,

We’re pleased to inform you that your manuscript has been judged scientifically suitable for publication and will be formally accepted for publication once it meets all outstanding technical requirements.

Kind regards,

Afzal Basha Shaik, Ph.D

Academic Editor

PLOS ONE
---

## [Editor Report · Acceptance letter]

17 Sep 2024

PONE-D-24-02007R2 

PLOS ONE

Dear Dr. Ahmed, 

I'm pleased to inform you that your manuscript has been deemed suitable for publication in PLOS ONE. Congratulations! Your manuscript is now being handed over to our production team.

Kind regards, 

on behalf of

Dr. Afzal Basha Shaik 

Academic Editor

PLOS ONE